# Denoising and iterative phase recovery reveal low-occupancy populations in protein crystals

Alisia Fadini [1,4] ✉, Virginia Apostolopoulou [2,3], Thomas J. Lane [2,3] ✉ & Jasper J. van Thor [1] ✉

Advances in structural biology increasingly focus on uncovering protein dynamics and transient macromolecular complexes. Such studies require modeling of low-occupancy species like time-evolving intermediates and bound ligands. In protein crystallography, difference maps that compare paired perturbed and reference datasets are a powerful way to identify and aid modeling of low-occupancy species. Current methods to generate difference maps, however, rely on manually tuned parameters and, when signals are weak due to low occupancy, can fail to extract clear, chemically interpretable signals. We address these issues, first by showing that negentropy – a measure of how different a signal looks from anticipated Gaussian noise – is an effective metric to assess difference map quality and can therefore be used to automatically determine difference map calculation parameters. Leveraging this, we apply total variation denoising, an image restoration technique that requires a choice of regularization parameter, to crystallographic difference maps. We show that total variation denoising improves map signal-to-noise and enables us to estimate the latent phase contribution of low-occupancy states. We implement this technology in an open-source Python package, METEOR. METEOR opens new possibilities, for time-resolved and ligand-screening crystallography especially, allowing detection of low-occupancy states that could not previously be resolved.

A growing number of cutting-edge macromolecular crystallography methods aim to resolve weakly populated states rather than the primary protein conformation present in a crystal. Mechanistic time-resolved studies at XFELs[1] and synchrotrons[2], high-throughput fragment screens[3], or the study of functional protein conformational changes upon the external perturbation of a crystal (using pH, temperature[4], or an external field[5]) are important examples. Because full conversion to a new state through a transient stimulus, like optical excitation[6] or substrate and ligand diffusion[7–10], is difficult to achieve, such techniques rely on the observation of small differences between crystallographic datasets. This complication can limit their application: photoactivated systems with low quantum yields, weakly-bound ligands, and bound small molecules used for hit-to-lead drug discovery all produce signals with strengths comparable to the experimental noise and will benefit from new analysis methods that aid interpretation.

Provided that perturbation-driven structural changes are not too large (i.e., the data remain isomorphous[11]), the calculation of difference electron density (DED) maps can be used to reveal changes in the electron density between a new dataset, which we refer to here as the derivative dataset, and an unperturbed native dataset. DED maps are used to identify regions of structural change and to extrapolate Fourier coefficients to which new coordinates can be refined[12–14]. Any derivative structure factor $\mathbf{F'}$ is a superposition of the initial native structure factor ($\mathbf{F}$) and the one related to the perturbed state of interest ($\mathbf{F^{pr}}$), where by perturbed state we mean the pure structure of interest, i.e., ligand-bound, time-activated, etc.:

$$\mathbf{F'} = f \times \mathbf{F^{pr}} + (1 - f) \times \mathbf{F}$$

here $f$ is the occupancy of the perturbed state and we indicate complex structure factors in bold. Combined with measurement noise, this partial occupancy means that dataset-dependent changes can be small, making DED map features difficult to interpret. Specific communities have developed different approaches to tackle this issue. For example, the Phenix suite[15]—commonly used in static structure comparisons such as ligand binding or point mutations—provides a simple utility for creating an

[1]Department of Life Sciences, Faculty of Natural Sciences, Imperial College London, London, UK. [2]Center for Free-Electron Laser Science CFEL, Deutsches Elektronen-Synchrotron DESY, Hamburg, Germany. [3]The Hamburg Centre for Ultrafast Imaging, Hamburg, Germany. [4]Present address: Department of Systems Biology, Columbia University, New York, NY, USA. ✉e-mail: af3659@columbia.edu; thomas.lane@desy.de; j.vanthor@imperial.ac.uk

isomorphous difference map from two sets of amplitudes. Time-resolved crystallographers use weights ($w$) that reduce the amplitudes of difference structure factors ($w$ x [$F'_{obs}$ - $F_{obs}$]) if they have large experimental error and are deemed to be outliers[12,16,17]. The Xtrapol8 program[12] makes a variety of such weighting schemes available to users, together with occupancy estimation strategies. One of the key weighting strategies implemented in Xtrapol8 is k-weighting, which adjusts the amplitude of each difference structure factor to reduce the influence of outliers. The weight applied to each difference structure factor amplitude ($\Delta F$) is calculated as:

$$w = [1 + (\sigma \Delta F^2 / <\sigma \Delta F^2>) + k(|\Delta F|^2 / <|\Delta F|^2>)]^{-1}$$

where $\sigma\_\Delta F$ is the uncertainty associated with a specific $\Delta F$, and k is a tunable scaling factor that controls how strongly large $|\Delta F|$ values—especially those with underestimated uncertainties—are down-weighted. Implementations using k = 1, k = 0, or intermediate values of k have all been applied in the literature[5,12,18]. For crystallographic fragment screening, the PanDDA suite[19] has introduced an objective procedure to identify density associated with partially-occupied ligands. PanDDA looks for regions with statistically significant excess density as compared to reference, ligand-free datasets. While both strategies can identify weak signals in DED maps, experimental maps frequently remain dominated by noise (Fig. 1). Moreover, while amplitude weighting schemes are powerful, they require the selection of appropriate weighting parameters, and this is left to the user's discretion. One common example, as mentioned above, is adjusting the outlier rejection term in k-weighting[12]. This manual intervention requires extensive trial-and-error that is based on visual inspection of maps and can introduce user bias.

Methods that further automate DED map estimation and increase the interpretability of low-occupancy density in an unbiased way are therefore desirable[20–22]; such tools will reduce user bias, provide faster feedback during time-resolved experiments or ligand screening campaigns, and allow accurate analysis of low-occupancy species where existing methods fail.

For this reason, we aim to develop methods that improve the signal-to-noise ratio in DED maps. To do so, we find it essential to establish an objective, quantitative, simple measure of map quality to compare different map generation strategies. Work in this direction is greatly aided by insights from other fields as well as open-source software that enables the development and distribution of new crystallographic analyses[15,23,24].

We therefore first focus on identifying a suitable and reliable statistic that reports the level of noise present in a difference density map. We evaluate statistics that measure deviations between the distribution of map voxel values and a Gaussian distribution as indicators of DED map quality, ultimately favoring negentropy. Map negentropy ($J(p_\rho)$) provides a useful metric for assessing how much structured, signal-like information is present in a difference map: a higher negentropy suggests the presence of interpretable features such as real electron density changes, rather than noise. Formally, negentropy is the difference in differential entropy between the distribution of DED map voxel values ($p_\rho$) and a Gaussian distribution with the same mean and variance ($p_{gauss}$) (under the assumption that individual voxels are identically and independently distributed)[25,26]:

$$J(p_\rho) = H(p_{gauss}) - H(p_\rho)$$

where the differential entropy $H(p_\rho)$ is defined as:

$$H(p_\rho) = - \int p_\rho(u) \log p_\rho(u) du$$

integrated over all voxels (u). Already routinely applied in independent component analysis[27–29] as a measure of non-Gaussianity, negentropy quantifies how far a signal deviates from Gaussian randomness. We show that maximizing negentropy is an effective approach for selecting parameters in models that aim to denoise DED maps.

We use this insight to propose a way to denoise DED maps through total variation (TV) minimization. In crystallographic DED maps, we expect a priori that the signals of interest should consist of local regions of smoothly-varying signal where atoms have moved, appeared, or disappeared. The rest of the map should be empty. When such signal is corrupted by additive white Gaussian noise, the noise contribution dominates at high spatial frequencies. TV minimization is an established technique in image processing[30] that aims to clarify the true signal by reducing these fluctuations. The TV is simply defined as the sum of the changes from each voxel to all neighboring voxels; minimizing this quantity suppresses small-scale noise while preserving important structural features. Applied as a density modification technique, TV denoising[31] thus produces a map that closely resembles the input map, but with flattened noise-dominated regions and preserved peaks or edges. TV denoising differs from commonly used low-pass filters, such as running averages or Gaussian filtering, by preserving sharp features (such as edges or peaks) while selectively suppressing small-scale fluctuations characteristic of noise. In contrast, low pass filters suppress high-frequency noise but blur sharp signal features[32]. TV denoising is therefore particularly well-suited to cases where enhancing sparse, localized signals is important, making it a promising approach for DED maps.

We apply TV denoising[31] to DED maps and use negentropy maximization to select a regularization parameter that effectively trades off between fidelity to the original map and a denoised result. For three distinct case studies, we show that a single pass of TV denoising boosts the signal-to-noise ratio of DED maps and improves interpretability. We also find that an iterative application of TV denoising can be used to estimate phases for the derivative **F'** dataset, with corresponding further improvement in map quality.

Finally, we demonstrate how TV-denoised maps can be used to obtain extrapolated maps of the perturbed state and also refine multi-state coordinates (combined reference and perturbed state) to an **F'** dataset, revealing protein backbone and sidechain rearrangements in a fragment-bound COVID-19 main protease ($M^{Pro}$) structure[33] that were not previously discernible.

We compile this analysis as an open source Python package, METEOR: Map Enhancement Tools for Ephemeral Occupancy Refinement. METEOR includes code to generate difference maps, select weighting parameters based on map negentropy, and apply an appropriate TV denoising protocol. Importantly, our observations on using map negentropy and TV denoising to improve difference density signals are compatible and stackable with existing methods. In particular, METEOR can complement existing occupancy estimation and event detection tools—such as PanDDA or Xtrapol8—by improving map interpretability prior to modeling. We expect this stacking to be especially useful in low-occupancy cases where difference map signal is weak.

## Results

### Negentropy reports on the interpretability of a difference map

We start by considering the voxel-value distribution for a difference map. In an ideal, error-free DED map where the atoms move large distances, we expect the distribution to be bimodal in nature, with a positive and a negative mode and otherwise zero-valued voxels (Fig. 1a). In the more realistic case where positive and negative peaks from different motions overlay and partially cancel out, the density distribution may no longer be strictly bimodal but should remain non-Gaussian (Fig. 1b, c). Noise, by the central limit theorem, can be expected to be an additive Gaussian contribution to the difference map signal. Empirically, we indeed observe that the distribution of voxel values for experimental maps is notably more Gaussian than that of synthetic maps calculated with no errors, suggesting that noise contributions dominate the former (Fig. 1d). We therefore postulate that looking for deviations from Gaussianity in the voxel-value distribution may enable maximization of DED features that are not noise.

Skewness, kurtosis, and negentropy are well-known measures for non-Gaussianity[27,28,34]. In addition to negentropy, defined in the previous section, skewness measures asymmetry in a distribution, while kurtosis describes the

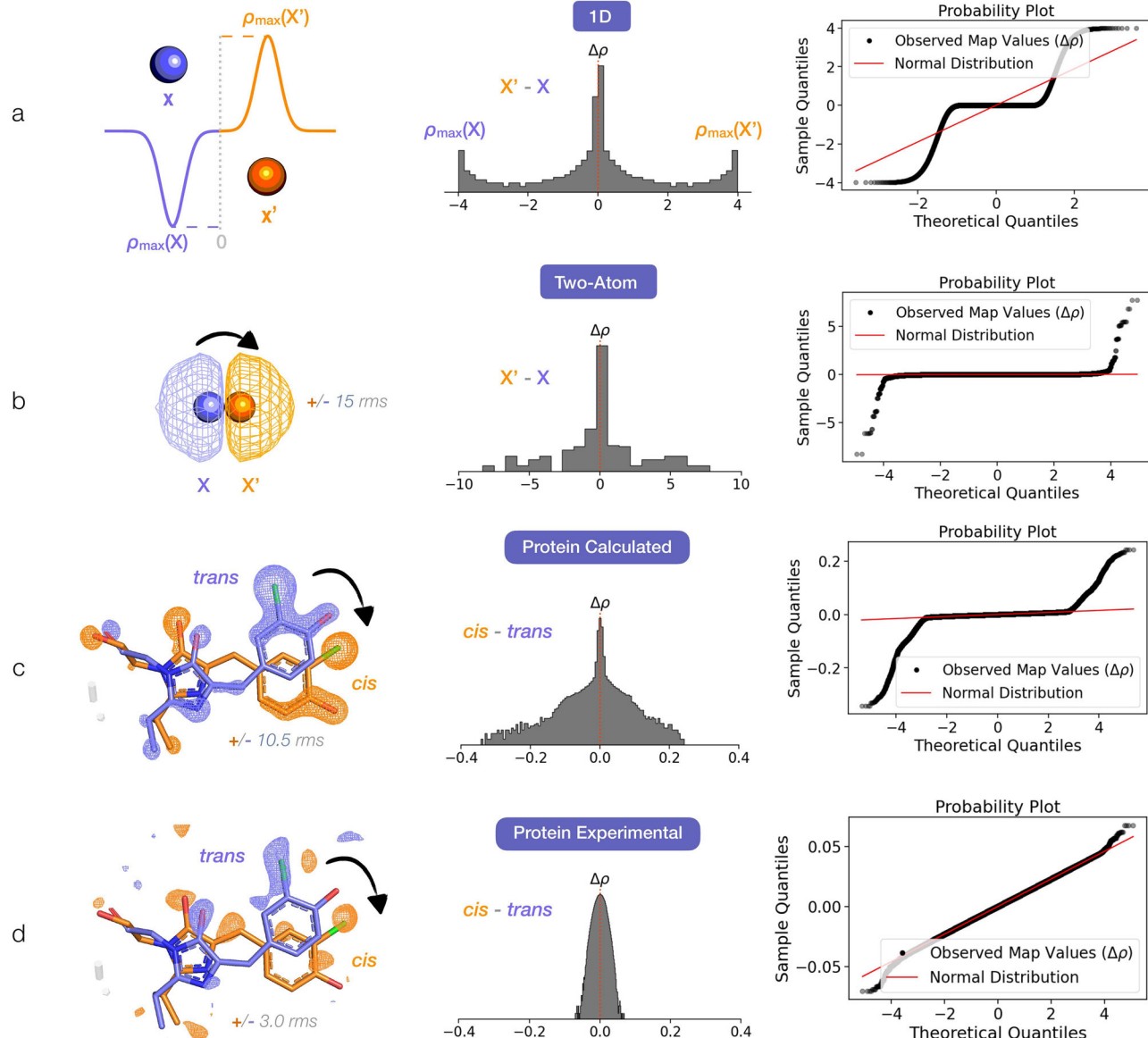

**Fig. 1 | The voxel distribution of a difference map reflects its signal-to-noise ratio.** **a** A 1D signal with positive and negative peaks at positions at X' and X respectively, can serve as a simple model for a difference map where an atom moves from X to X'. The corresponding distribution for the difference map voxel values ($\Delta\rho$) is non-Gaussian, with a mode at 0 and two modes at $\rho_{max}(X)$ and $\rho_{max}(X')$ (the histogram is plotted with a log-scale on the $y$-axis). On the right column, the normal probability plot compares the observed map voxel value distribution to what would be expected if the data followed a perfect Gaussian (normal) distribution: the observed data (sample quantiles) are ordered and plotted against the expected values of the ordered statistics for a sample from a standard normal distribution (mean 0, variance 1) of the same size as the data (theoretical quantiles). If the observed values closely follow the red diagonal line, this indicates that the data are approximately Gaussian. Deviations from this straight line indicate deviations from a Gaussian distribution. **b** A more realistic case is that of a carbon atom translating from X to X' in three-dimensional space. Here, we show the case where the positive and negative electron

density signals from the displacement of the atom overlay and partially cancel. The resulting difference map voxel value distribution ($\Delta\rho$) is no longer strictly bimodal but remains non-Gaussian. **c** Deviation from a Gaussian distribution for the histogram of voxel values is also noticeable for the calculated difference map of a known light-induced protein structural change: we display a noise-free synthetic difference electron density (DED) map for the 100 ps trans-to-cis photoisomerization of the Cl-rsEGFP2 protein chromophore (PDB ID 8A6G). To match experimental data31, the map is calculated by converting 13% of the reference dark trans population to cis. As the dark state already contains 14% of the cis species, the final state (in orange here) is at 27% cis occupancy. We show the calculated difference map at a contour level that is comparable to the experimental map shown below in (**d**). In contrast to the calculated map, the experimental map35 is dominated by Gaussian noise, even after error-based amplitude k-weighting. This is apparent from the histogram and probability plot of the map voxel value distribution shown on the right.

"tailedness" or sharpness of its peak. To investigate the suitability of these statistics as indicators of difference map quality, we extend the examples from Fig. 1a–c by adding Gaussian noise to their calculated real-space signal (see Fig. S1). We introduce increasing levels of noise and find that negentropy decreases monotonically with the addition of noise and is the most robust in its behavior when compared to skewness and kurtosis (Fig. S1). On

the basis of this test, we proceed with the proposal that negentropy could be a useful metric to evaluate the signal-to-noise ratio in DED maps.

We observe, first of all, that by maximizing the negentropy, we can optimize parameters in the commonly used k-weighting[12,17] amplitude modification scheme (Fig. S2). We note, however, that, even after k-weighting, the voxel value distribution for experimental difference maps

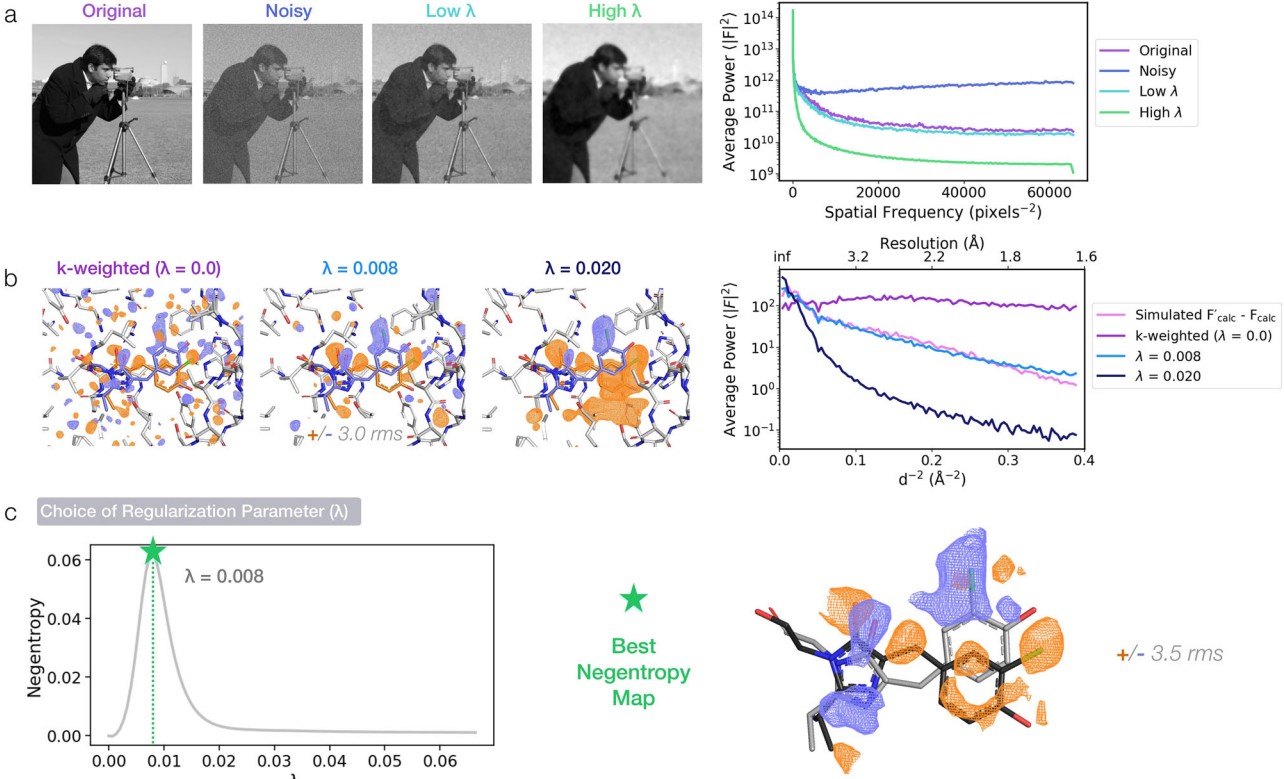

**Fig. 2 | Total variation (TV) minimization can effectively denoise experimental DED maps. a** TV denoising is used in signal processing to remove spurious noise from data. This is exemplified here by artificially adding white noise (sampled from a normal distribution with a standard deviation ≈10% of the maximum pixel value in the image) to the cameraman image and applying Chambolle's total variation minimization algorithm to retrieve the underlying signal32. A regularization parameter λ determines the degree of denoising: a value of λ that is too high will produce an image that is overly "smoothed" compared to the original. On the right, the power spectrum for each image illustrates the effect of the added noise at different spatial frequencies and of subsequent TV denoising to recover signal. **b** TV denoising using two different manually chosen values of λ is shown for the k-weighted Cl-rsEGFP2 trans-to-cis photoisomerization DED map, along with the original experimental map (λ = 0.0). As for the 2D images in (**a**), power spectra show that, while the starting experimental map contains little signal, an appropriately regularized denoising yields map coefficients that approach those for the simulated map displayed in Fig. 1c. For visual comparison, the different map spectra are presented on a common scale. **c** To refine an appropriate value of λ, we denoise the experimental Cl-rsEGFP2 map for a range of regularization levels (0 ≤ λ ≤ 0.08) and identify the λ value that maximizes map negentropy.

often remains markedly Gaussian (see below) and hypothesize that further denoising could yield important improvements.

## Total variation denoising facilitates interpretation of weak signals in difference maps

One such case where even the amplitude k-weighted DED map is extremely noisy is the 100 ps *trans*-to-*cis* photoisomerization of the Cl-rsEGFP2 protein chromophore reported in Fadini et al.[35] (PDB ID 8A6G) and shown in Fig. 1d. This dataset captures a spatially localized, light-induced change where the *cis* photoproduct coordinates are well-known from ground state synchrotron structures[36] and is therefore an excellent example for methods development. We use it here to evaluate how TV denoising can assist in the interpretation of noisy maps.

Our hypothesis is that TV denoising should remove unwanted high-frequency noise, while preserving the signal features of difference density that vary more smoothly. A necessary step in TV denoising is to choose the degree of denoising, dictated by a regularization parameter, λ; this controls the trade-off between denoising strength and how closely the result matches the original map (see Supplementary Note S1 for definition and formulation). To gain intuition into the effect of different choices of regularization parameter, we compare the original k-weighted Cl-rsEGFP2 DED map to two denoised maps, where we set λ manually based on visual interpretability of the resulting DED map (Fig. 2). TV denoising with a moderate level of regularization (λ = 0.008) produces a map with stronger signals on the protein chromophore and removes much of the noise from the original

map. An overly-aggressive denoising (λ = 0.02), leads to a less interpretable map (Fig. 2b). To support this subjective assessment, we plot the power spectra of the DED maps. The power spectrum of a difference map describes how the signal is distributed across spatial frequencies (resolution shells), with higher frequencies corresponding to finer structural detail. By comparing power spectra before and after denoising, we can assess whether the process has effectively suppressed high-frequency noise while preserving meaningful signal. The power spectra show that the moderate value of λ (0.008) restores signal and recovers the expected resolution-dependent behavior of a noise-free synthetic map, while the higher λ (0.02) leads to a power spectrum that deviates from the "ground truth" spectrum computed from the synthetic example.

We conclude that TV denoising is a promising method for suppressing noise in DED maps, but would benefit from a robust, objective, and automated choice of the regularization parameter. To test whether maximizing the negentropy of DED maps as a function of λ could effectively serve this purpose, we simulate a *trans*-to-*cis* difference map that includes additive noise and track map negentropy while screening a range of λ values (Fig. S3). For a simple model we add random noise to the difference density in real space. We observe that the value of λ that maximizes negentropy closely matches the value that maximizes the real-space correlation coefficient between the denoised map and the noise-free synthetic map (Fig. S3c). We therefore propose that the regularization parameter λ can be chosen by finding the value that generates the highest negentropy difference map (Fig. 2c).

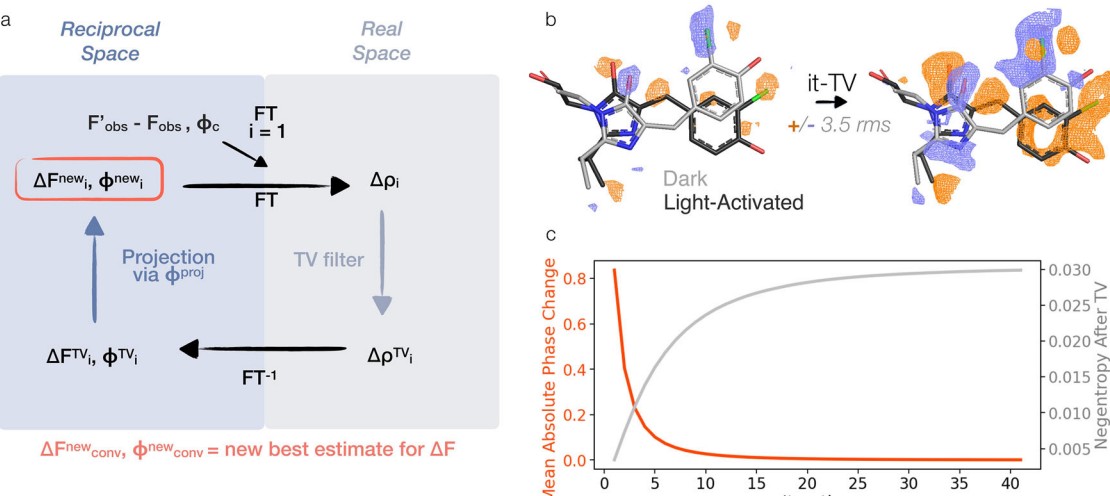

**Fig. 3 | An iterative TV minimization algorithm estimates the phases of the structure factor contribution from low occupancy states. a** The unknown perturbed state phases are estimated through an iterative procedure (it-TV, Fig. S6). The initial difference map is computed from the observed derivative and native amplitudes (F'$_{obs}$ and F$_{obs}$) and the phases from the reference state model ($\phi_c$). A step of difference density ($\Delta\rho$) TV denoising is followed by an inverse Fourier transform (FT) to reciprocal space. We then project the **ΔF$^{TV}$** set onto the phase circle with magnitude F'$_{obs}$ (see also Fig. S6) as a way of finding a new phase estimate for **F'**. The new phases are used in the next iteration. Iterations are run until convergence (see "Methods"). **b** The experimental Cl-rsEGFP2 map is shown before and after it-TV (in purple and orange for negative and positive density, respectively, ±3.5 rms), together with the absolute mean phase change and negentropy values for the difference maps generated at each iteration in (**c**). The absolute mean phase change at each iteration is plotted in red. The negentropy as a function of iteration is plotted in gray.

While the TV denoising step modifies the Fourier amplitudes of the map, it also alters the Fourier phases (Fig. S4). This is notable, as the phases traditionally used in DED maps are approximate: the **F'** phases, which have a one-to-one correspondence to the perturbed state phases are unknown. These phases are usually approximated by the native dataset phases coming from a well-characterized reference model ($\phi_c$)[11]. This assumption, however, approximately halves the signal-to-noise ratio in the corresponding map as compared to what would be obtained if the perturbed state phases were known[37], an undesirable effect, particularly when the perturbed signal is weakened by partial occupancy. Our observations suggest that TV denoising, which modifies the phases, may partially correct for this approximation.

This insight points to the potential use of TV denoising for iterative phase improvement, employing a cycle similar to that used for solvent flattening in crystallography[38] or phase retrieval in coherent diffractive imaging[39,40]. Inspired by this prior work, we use the set of TV-denoised difference structure factors to better estimate the phases for the derivative dataset, **F'**, through an iterative algorithm that we name iterative-TV (it-TV) (see "Methods" and Fig. 3). We again first validate the it-TV approach on a synthetic noisy map, where the noise-free, ground truth map is known (Supplementary Note S2 and Fig. S3). In Fig. 3, we show the experimental Cl-rsEGFP2 map before and after it-TV, together with the absolute mean phase change and negentropy values for the difference maps generated at each iteration. Map negentropy reaches a stable positive value, having started below $10^{-4}$ for the original experimental map, and the final it-TV map contains notably higher signal-to-noise difference signal on the protein chromophore. This first result indicates that application of TV denoising as a density modification approach could find improved phase estimates for low occupancy states in DED maps.

**Test cases demonstrate the power of negentropy-guided TV denoising to recover time-resolved and ligand-binding signals**
To benchmark the single-pass and iterative TV denoising techniques, we select three distinct science cases:

- The 100 ps time-resolved crystallography Cl-rsEGFP2 dataset from Fadini et al.[35] (PDB ID 8A6G).
- The example of M$^{pro}$ bound to tegafur (fragment SW7-401, PDB ID: 7AWR), identified as a potential binder to an allosteric site with a modeled occupancy of 0.50[33].
- The example of M$^{pro}$ bound to a small electrophilic fragment (U1G, PDB ID: 5RGO) with a modeled occupancy of 0.42[41].

For Cl-rsEGFP2 and the M$^{pro}$-tegafur complex, we compute difference maps using the sets of observed amplitudes for the available derivative and native datasets (F'$_{obs}$ - F$_{obs}$). For the M$^{pro}$-U1G complex, we illustrate the case where the difference map is computed between the observed amplitudes and the ones calculated from the model (F$_{obs}$ - F$_{calc}$), when a reference native dataset is not available. Both M$^{pro}$ structures score very poorly for ligand goodness of fit to experimental data in their respective PDB depositions[42], suggesting that the ligand density could benefit from further improvement.

The first two columns in Fig. 4 show the effect of TV denoising on DED maps for our three test cases: (a) Cl-rsEGFP2, (b) M$^{pro}$-tegafur complex, (c) M$^{pro}$-U1G complex. Features like the outline of the chlorophenolate ring from the Cl-rsEGFP2 *cis* chromophore are more easily identifiable within the protein structure and more chemically interpretable: at ±3.5 rms, a large negative feature is visible on the imidazolinone ring after TV denoising, and the positive density on the *cis* chlorophenolate ring extends to four carbons and the hydroxyl group. Relevant signals on the chromophore appear significantly stronger following TV denoising. For instance, the positive and negative peaks on the chlorine atom reach ±11 rms in the TV-denoised map, compared to ±5 rms in the original map. For the M$^{pro}$-U1G complex, the denoising removes a large portion of uninterpretable density next to the ligand. In each case, visual improvements in the denoised maps are supported by increased negentropy for the voxel value distributions. In the Cl-rsEGFP2 case, the map negentropy increases from less than $10^{-4}$ to 0.059 (for comparison, the Cl-rsEGFP2 noise-free synthetic map shown in Fig. 1c is associated with a negentropy value of 0.35). An analysis of the changes

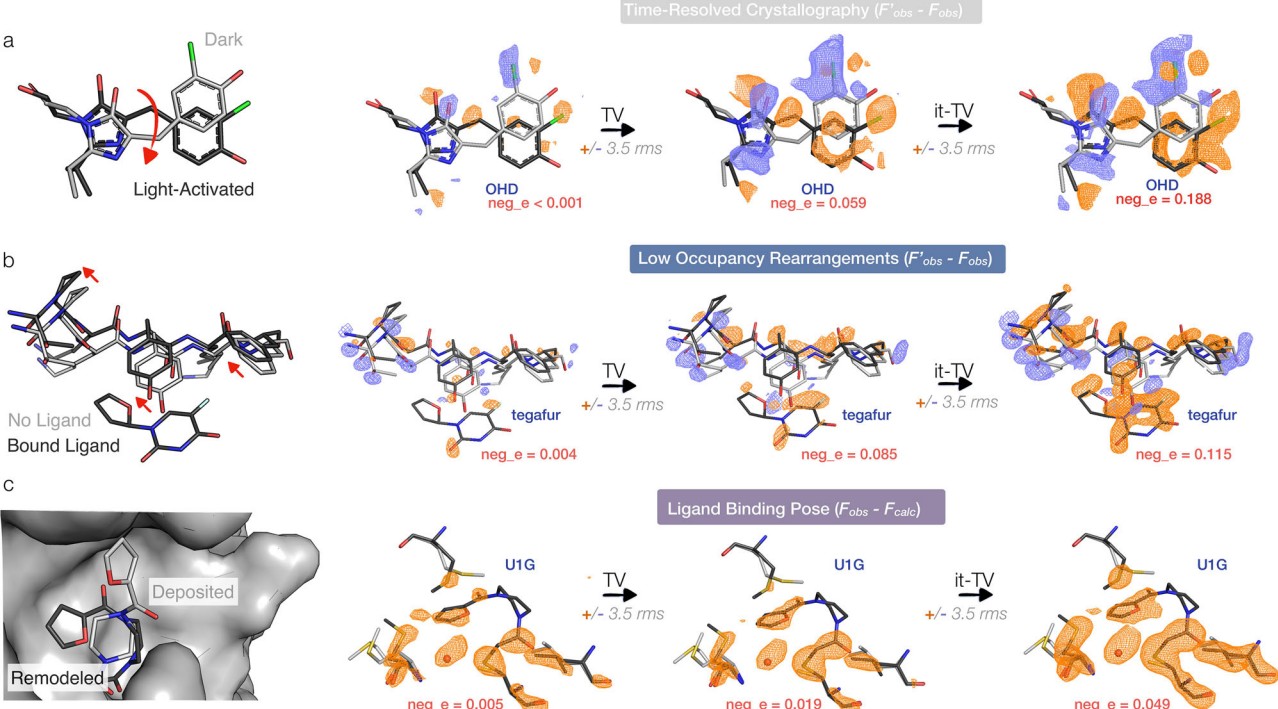

**Fig. 4 | Test cases demonstrate the power of negentropy-guided TV denoising to recover time-resolved and ligand-binding signals.** We show single-pass TV denoising and iterative-TV maps with their associated negentropy values for our three test datasets: the 100 ps photoisomerization of the OHD chromophore in Cl-rsEGFP2 protein (PDB ID 8A6G) (**a**), the example of M$^{pro}$ bound to tegafur (PDB ID 7AWR), which was identified as a potential binder to an allosteric site with a modeled occupancy of 0.50 (**b**), and the example of the M$^{pro}$-U1G complex (PDB ID 5RGO), with a modeled fragment occupancy of 0.42 (**c**). Reference state structures are shown in gray, while structures that we refined to the perturbed dataset are shown in black. Ligand outlines become stronger and more chemically interpretable for all three test cases. For the M$^{pro}$-tegafur complex, there are strong signals for rearrangements of side chains and backbone atoms near the ligand binding pocket.

For the M$^{pro}$-U1G complex, the it-TV map suggests an alternative fragment pose. The largest gains from the denoising step occur when the initial map is close to being normally distributed (as for the signal on the OHD chromophore in Cl-rsEGFP2 or in the M$^{pro}$-tegafur complex). We show maps at ±3.5 rms, which we find is an appropriate rms cutoff to highlight signal for these examples. However, because TV-denoised maps are intentionally non-Gaussian, the second moment (rms) alone does not fully capture the distribution of voxel values. It's therefore important to keep in mind that common visualization thresholds, like ±3 rms, may not directly relate when trying to compare denoised and standard maps. The improved interpretability seen here is further supported by the quantitative distribution analysis shown in Figs. S4 and S5.

introduced by the denoising step reveals that the largest modifications involve high-resolution structure factor amplitudes and phases (Fig. S4). Finally, supporting the statement that the denoising step removes noise from the maps, the histograms and probability plots show that voxel value distributions are less Gaussian after TV denoising (Fig. S5).

The it-TV maps for our three test cases can be found in the third column in Fig. 4. These show clear differences that can be explicitly assigned to molecular structure. They are all characterized by an increased negentropy and less Gaussian voxel-value distributions compared to their respective originals and single-pass TV maps (Fig. S7). Note, for example, that the Cl-rsEGFP2 it-TV map in Fig. 4 shows a clear outline of the chlorophenolate ring from the *cis* photoproduct, even though *cis* phases calculated from an atomic model are never introduced. Similarly, in the M$^{pro}$-tegafur complex map, an almost-complete outline of the fragment's fluorouracil ring can be seen at 3.5 rms without ever using the fragment structure in the analysis. For the M$^{pro}$-U1G F$_{obs}$–F$_{calc}$ case, it-TV increases the strength of positive signals on the side chain atoms and ligand. Once more, the strongest modifications by the algorithm occur in higher resolution shells (Fig. S4), effectively recovering the underlying high resolution signal from the starting map.

### Denoised difference maps can guide the refinement of new coordinates

Because DED maps do not require a model of the perturbed state, they provide unbiased and immediate information about the structural changes

that occur after a perturbation, such as ligand binding or light-induced protein motion. This is invaluable to understand if the experiment has been successful and propose models that explain the results. The ultimate aim of most crystallographic experiments, however, is to interpret these changes chemically i.e., in terms of the positions of atoms and bonds. This requires an atomic model of the perturbed state, which can be refined using an extrapolated map. The extrapolated map should ideally reveal the electron density of the perturbed state without contributions from the reference state, making it easier to identify perturbation-specific structural change[12,18].

To produce an extrapolated map, an accurate estimate of the perturbed state occupancy is essential: once an estimate for the occupancy is known, the extrapolated map can be computed by performing an appropriate addition in real space (between a reference map and a difference map, as done by PanDDA) or in reciprocal space (between reference structure factors and difference structure factors, as done by Xtrapol8). We use the real-space approach here to illustrate how TV denoised difference maps can improve interpretability.

During this extrapolation procedure, TV-denoised maps are useful for identifying the region of the difference signal to focus on for initial occupancy estimation (Fig. S8). They can also improve the extrapolated map resolution: Fig. 5a displays the extrapolated map obtained by real-space addition between the reference M$^{pro}$ state (PDB ID 7AR6) map and the M$^{pro}$-tegafur it-TV map (see "Methods"). The map shows a clear shift of the backbone and sidechains in the ligand-binding pocket and can be used to refine new atomic coordinates for the ligand-bound state. This low-

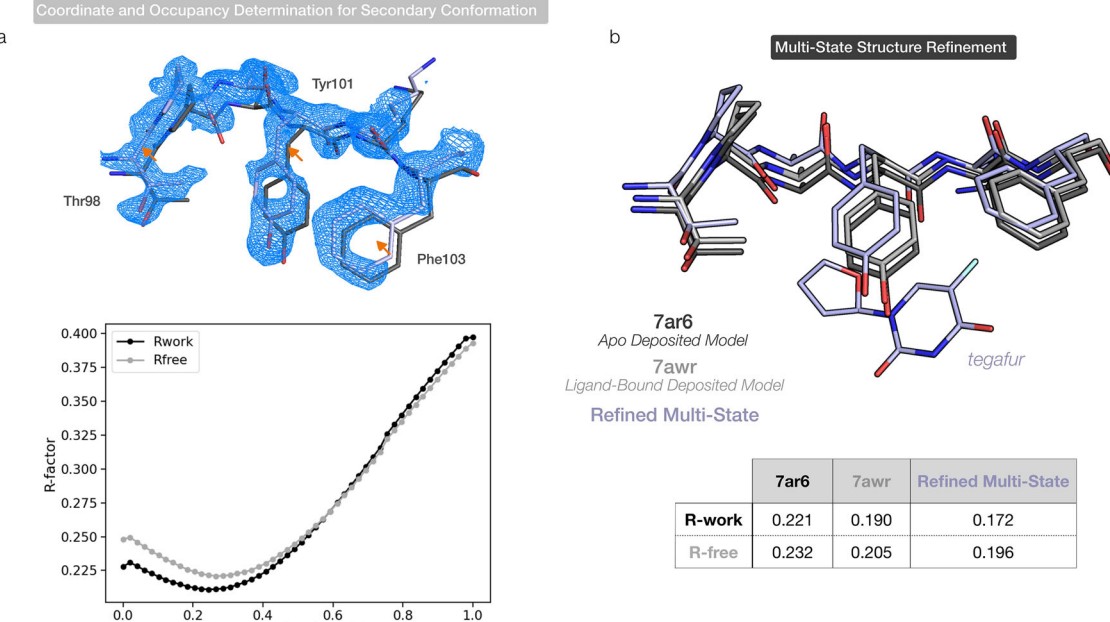

**Fig. 5 | Denoised difference maps can guide the refinement of new coordinates and uncover low-occupancy conformations. a** Extrapolated map for the M^pro-tegafur complex. This is obtained through addition of the it-TV map shown in Fig. 4 to the 2mF_o-DF_c map from the reference state (PDB ID 7AR6, black). We highlight the rearrangement of the backbone and Thr98/Tyr101/Phr103 sidechain atoms close to the ligand binding pocket. We refine coordinates for the bound state to this extrapolated map (light blue) and screen R-factor values to initiate the occupancy of

the ligand-bound state in a multi-state model, choosing an occupancy of 0.29 for initial refinement. **b** Binding pocket coordinates from the deposited ligand bound (PDB ID 7AWR) and unbound (PDB ID 7AR6) models and the multi-state model generated here are shown. The table reports final refinement R-factors for the deposited ligand bound and unbound models and the multi-state model when compared to the ligand-bound dataset. For the refined multi-state model, the reference chain is overlaid with the 7AR6 model.

occupancy conformation was not observable in the original PanDDA event map (Fig. S10a). We can refine a multi-state model of ligand bound and unbound states directly to the ligand-bound dataset. Figure 5b compares the model from this refinement with the deposited structure for the ligand-bound state (PDB ID 7AWR)[33]. We note that refining a single model to the ligand-bound dataset results in atomic positions that are effectively an average between the two states in the multi-state model and are less representative of the underlying system.

An additional use for it-TV maps in the process of modeling perturbed-state coordinates is to suggest an initial ligand pose for refinement. We use the it-TV denoised map for the M^pro-U1G complex to manually remodel the ligand binding pose (Fig. 4) and improve model fit to the data compared to both the deposited structure and a simple re-refinement of the deposited coordinates (Fig. S11).

## Discussion

Exciting applications of crystallography, such as time-resolved mechanistic studies or high-throughput fragment screens, struggle with the challenge of analyzing signal from low-occupancy species, which is at the same level as the noise in the data. Therefore, methods to measure and suppress noise in these datasets can unlock scientific opportunities.

To this end, we find negentropy to be an effective reporter of difference map quality when seeking optimal amplitude weighting and denoising parameters. Negentropy maximization alone, however, cannot be used for map denoising without other restrictions to the procedure. Consider a putative DED map with half of the voxels randomly set to +x and half to −x. Such a map will have a high negentropy, but no information content. Instead, negentropy, which reports how noise-like a signal is, should be thought of as a measure of the signal-to-noise ratio in a map. It can tell if the map contains signal, but not if that signal is faithful to any crystallographic or biological reality. Therefore, negentropy should only be used to evaluate competing DED maps generated by methods that can maintain fidelity to

the original crystallographic data. As examples here, we show the need to choose the k parameter in k-weighting or the λ parameter in TV denoising, where negentropy maximization replaces manual selection, improving automation and reducing user bias.

We hypothesized that by extending single-pass total variation denoising (single-TV) into an iterative map denoising algorithm (it-TV), it would be possible—at least in theory—to recover phase information and recover a fraction of the signal-to-noise loss relative to traditional F'_obs−F_obs maps that rely on the native-phase approximation. Across all three test cases, the it-TV output maps exhibit greater chemical interpretability and higher negentropy than those produced by a single pass of TV denoising, suggesting that iterative denoising adds significant value. Our current it-TV implementation, based on the Gerchberg-Saxton iterative projection algorithm[43], achieves this improvement effectively; while in principle such algorithms can become trapped in local minima, we have not observed clear failure cases in our test data so far. Nonetheless, this could remain a consideration for more challenging datasets and motivates future work on more robust or regularized update schemes, such as Feinup's hybrid-input output or its descendants[39], to enhance it-TV robustness against noise-induced errors. Like solvent flattening and phase retrieval, the it-TV method does not involve an atomic model or user intervention and thus avoids model-driven phase bias. That said, it could be possible for phase modification to introduce spurious features that might be biologically interpreted. To counter this possibility, we recommend comparing it-TV maps with both single-pass TV maps and unprocessed maps. When denoising is successful, these maps should be consistent and not in contradiction.

The largest gains from our TV denoising tests occur for the Cl-rsEGFP2 F'_obs-F_obs map, which is expected to benefit the most from the replacement of the single-phase approximation in it-TV and also represents the lowest occupancy example. We therefore anticipate that our denoising methods will be especially useful in pushing the boundary for interpreting datasets from low-occupancy time-resolved studies or weakly bound

ligands. It is important to note that METEOR, as described throughout this work, is designed to address Gaussian noise in DED maps and does not inherently correct for systematic errors common in crystallographic datasets, such as scaling artifacts or anisotropy. To improve the reliability of denoised maps, we recommend applying tools specifically developed to correct for such systematic effects as part of standard preprocessing prior to TV denoising. For example, AIMLESS[44] adjusts the measured reflection intensities to correct for systematic errors and variations—such as changes in beam intensity, or detector sensitivity—so that all measurements are placed on a common, consistent scale for accurate merging and STARANISO[45] applies direction-dependent scaling corrections and resolution cutoffs to address anisotropic diffraction; SCALEIT offers local scaling procedures that mitigate systematic variation in structure factor amplitudes. These approaches are well-established in difference map workflows and are complementary to the denoising strategies we present here.

A further note of caution relates to the behavior of TV denoising in the absence of signal. In such cases, TV minimization may still produce localized, structured features due to the well-known "staircasing" effect[31,46]. This occurs because the method encourages piecewise-smooth output and may interpret random fluctuations as edges when no real signal is present. To help identify such false positives, METEOR issues a warning if the optimized regularization parameter λ exceeds the empirical threshold of λ > 0.1, which we have observed in signal-free cases and is a symptom of over-denoising. We also recommend users examine the denoised features: TV denoising will not reveal signals that were previously unobservable; rather, it improves interpretability by suppressing background noise, allowing real but weak features—i.e., previously visible only at low contour levels—to emerge more clearly. Future developments in DED map denoising could focus on a more holistic approach. For example, recent advances in deep learning, particularly convolutional neural networks and diffusion-based architectures, have shown strong performance in image restoration tasks[47–49] and are beginning to find applications in structural biology, including electron density map enhancement[50–52]. These models can learn complex, data-driven priors and may be capable of distinguishing subtle signal from structured noise or artifacts even in low signal-to-noise regimes. However, current deep learning approaches often require large, well-annotated training datasets, which, for example, are not readily available when dealing with DED maps. By contrast, our TV-based denoising framework is data-agnostic and easily deployable, requiring no training phase. This makes it equally suited to both fragment screening datasets and time-resolved experiments, including those that capture rare or previously unobserved conformational changes. TV denoising is fast, computationally inexpensive (on a single core, a typical TV run will complete in around 1 min, while it-TV will complete in around 3 min—Fig. S12), and can be run from most workstations without the need to train an additional set of parameters. An additional point is that there is scope for future DED map denoising to incorporate more sophisticated models for perturbed-state phases or for the errors introduced by the experimental measurement and imperfect reference-state coordinates[53,54].

The field of macromolecular crystallography is moving towards novel and technically ambitious data collection strategies: cutting-edge experiments are now aimed at capturing transient reaction intermediates, leveraging high-throughput facilities for small molecule screens, and studying the conformational variability that underlies protein dynamics. Analysis methods need to match these advances with increased sensitivity, robustness to noise, and improved automation. The negentropy metric and TV denoising techniques reduce human bias and noise in the generation of difference density maps, as well as enhance signal by proposing new phase estimates for low-occupancy species. We show that our analysis particularly improves data interpretation in cases that struggle with weaker signals. For these reasons, we believe such treatment of difference maps could unlock the study of minorly-populated states that are currently discarded in fragment screens or not attempted in time-resolved studies. The use of negentropy as

a way of systematically scoring difference maps should also be useful for quick decision-making during online experiments and for processing large crystallographic datasets from high-throughput beamlines. Beyond crystallography, we anticipate that this general framework could prove useful in related structural techniques such as cryo-EM, where weak, low-occupancy signals in maps are also a limiting factor due to partial ligand binding or conformational variability.

## Methods

### Total variation denoising

To conduct TV denoising on a difference map, we employ Chambolle's algorithm[31] implemented in scikit-image[55]. TV denoising is performed in real-space on difference map arrays. To fit the λ parameter, map negentropy is minimized using a golden section search[56,57]. For the skimage.restoration.denoise_tv_chambolle function, the tolerance for the stop criterions is set to $5 \times 10^{-8}$ and the maximum number of iterations to 50.

We have observed that schemes which modify difference structure factor amplitudes affect the total power of the related DED map. For example, both k-weighting and subsequent TV denoising result in a reduction in power compared to a simple Fo-Fo map. We show this is true for the Cl-rsEGFP2 data used throughout our work in Fig. S13. It is important to account for this change in overall power when using a modified set of difference structure factors (k-weighted, denoised, or both) for occupancy extrapolation. To ensure that METEOR maps are on the same scale as the original inputs, we rescale weighted or denoised maps back to the corresponding original difference map. To do this, we apply a strategy analogous to gain normalization in signal processing, where a denoised signal is rescaled to match the expected power of the original input in signal-dominated frequency bands. For example, in audio signal processing, it is common practice to normalize audio signal after denoising to adjust the amplitude of the cleaned signal to a desired level (this ensures consistency in loudness and prevents issues like clipping or distortion). In the case of difference maps, we reasoned that the signal-rich region of the map that will be least affected by the TV filtering process is the low-resolution regime. We therefore scale METEOR maps back to original input maps using low-resolution reflections (Fig. S13). The default is to use reflections that are lower than 1.5 Å from the stated resolution cutoff. We additionally include a comparison between the application of TV denoising and two other common types of filtering procedures to DED maps in Fig. S14.

### Iterative total variation denoising algorithm

Native ($F_{obs}$) and derivative ($F'_{obs}$) state amplitudes are provided as input, together with a reference model. A difference map is computed using the phases calculated from the reference model and the difference structure factor amplitudes ($F'_{obs}-F_{obs}$). This starting difference map is denoised using Chambolle's TV algorithm with a λ value optimized via negentropy. The resulting denoised map is inverse Fourier-transformed to obtain a set of complex structure factor differences, $\Delta F^{TV}$. We then project these onto the constraint circle defined by the derivative-state amplitudes ($|F'_{obs}|$) to estimate new phases $\phi^{proj}$ (Fig. S6). This projection aims to provide improved phase estimates for the derivative dataset. Using these refined amplitudes and projected phases, we compute a new difference map, apply TV denoising again, and repeat this procedure until convergence—defined as when the mean absolute phase change between iterations drops below $10^{-3}$ degrees. By default, METEOR then applies k-weighting and a final round of TV denoising as post-processing.

### Perturbed state extrapolation

To demonstrate how TV-denoised maps can be used in the process of extrapolating perturbed-state density, we generate two different types of extrapolated maps (Figs. 5 and S8–S10), described below. First, to estimate a perturbed state occupancy, we re-implement the background subtraction strategy used by PanDDA[19]. This procedure involves identifying a local region of change caused by the perturbation and finding an occupancy value

that can produce a map that maximally differs from its reference in such localized region but is similar to the reference elsewhere.

Extrapolation for perturbed state structure factors ($\mathbf{F^{pr}}$) or density ($\rho^{pr}$) can be carried out in real or reciprocal space:

$$\text{Reciprocal} : \mathbf{F^{pr}} = F_{obs} + \alpha^{-1} w \Delta F \times e^{i\phi_c}$$

$$\text{Real} : \rho_{pr} = \rho^{ref} + \alpha^{-1} \Delta \rho$$

where $\Delta F = F'_{obs} - F_{obs}$, $w$ represents error-based amplitude weights, $\phi_c$ are the calculated phases from the reference model, and $\alpha$ is related to the true perturbed state occupancy ($f$) by $f = 2\alpha$ when the constant phase approximation is used[37]. To estimate $\alpha$ (with final choice written $\hat{\alpha}$), the strategy from PanDDA finds the value that maximizes the difference in Pearson correlation coefficient (calculated from the reference state and the extrapolated map) between the entire protein and a specific local region of change. We showcase this for our photoisomerization example in Fig. S8a, where the local region is set as a sphere of 5 Å centered around the chromophore double bond. The entire protein is defined as a global region after a solvent mask is applied. For a range of values of $0 < \alpha < 1$, the Pearson correlation coefficient between the respective $\mathbf{F^{pr}}/\rho^{pr}$ map and the map obtained from the reference model is computed. The value that maximizes the difference between these two correlation coefficients is chosen as $\hat{\alpha}$ and the corresponding map is saved (Fig. S8b).

For the $M^{pro}$-tegafur complex, we use the higher-negentropy it-TV map as $\Delta \rho$ for real-space extrapolation and show the extrapolated result in Fig. 5. We also carry out a reciprocal space extrapolation, which does not use new phases from either single-TV or it-TV but is guided by the denoised maps in the choice of local region. The corresponding map is shown in Fig. S10b. Figure S9 contains the analysis to estimate $\hat{\alpha}$ for these two extrapolation types.

## Multi-state model refinement

As outlined by Pearce et al.[58], we combine the models through the use of alternate conformers, we constrain the occupancy of the atoms within each state to be fixed, and we set the sum of the occupancies between states to be equal to 1. As a starting occupancy for the ligand-bound state, we choose the value that minimizes R-free (Fig. 5b), and proceed to refine B-factors and coordinates with phenix.refine[59]. For the R-factors reported in the table in Fig. 5b, the anisotropic B-factors are removed from the originally deposited models (PDB ID 7AR6/7AWR) and the B-factors are re-refined with phenix.refine. This ensures a consistent procedure to compare the metrics obtained with the multi-state model refinement.

## Reporting summary

Further information on research design is available in the Nature Portfolio Reporting Summary linked to this article.

## Data availability

Map files from TV denoising examples shown are available at: https://zenodo.org/records/15800905[60].

## Code availability

Negentropy-driven k-weighting, single pass TV denoising, and the it-TV algorithm are implemented as executables in the METEOR package, requiring initial MTZ files and a reference model as input. METEOR is written in Python and depends on the GEMMI[22], reciprocalspaceship[24], numpy[61], scikit-learn[62], and pandas[63] packages. The code is available at: https://github.com/rs-station/meteor, https://doi.org/10.5281/zenodo.17103086[64].

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

## Acknowledgements
A.F. and J.J.v.T. acknowledge funding from the Imperial College President's PhD Scholarship and the Biotechnology and Biological Sciences Research Council (BBSRC) (BB/P00752X/1). T.J.L. acknowledges the support of the Helmholtz Association through a YIG award. V.A. and T.J.L. were supported by the Cluster of Excellence "Advanced Imaging of Matter," Deutsche Forschungsgemeinschaft (DFG), EXC 2056, project ID 390715994. We acknowledge Nick Pearce for discussion on the background subtraction estimation. We would also like to deeply thank Kevin Dalton for scientific exchange and for his help integrating METEOR into the *Reciprocal Space Station* platform.

## Author contributions
Conceptualization: A.F., T.J.L.; Methodology: A.F., T.J.L.; Investigation: A.F., T.J.L., V.A.; Visualization: A.F., T.J.L., V.A., J.J.v.T.; Funding acquisition: J.J.v.T.; Writing: A.F., T.J.L., V.A., J.J.v.T.

## Competing interests

T.J.L. is a shareholder of CHARM Therapeutics. All other authors declare no competing interests.
