## [Transparent Peer Review file · Communications Biology]

Denoising and Iterative Phase Recovery Reveal Low-Occupancy Populations in Protein Crystals

Corresponding Author: Professor Jasper van Thor

Version 0:

Reviewer comments:

Reviewer #1

(Remarks to the Author)

Report on:

Denoising Reveals Low-Occupancy Populations in Protein Crystals

Alisia Fadini, Virginia Apostolopoulou, Thomas J. Lane, Jasper J. van Thor

This article describes an approach to improving the signal-to-noise ratio in difference Fourier electron density maps so that weak difference signals may be resolved with improved confidence. Although difference Fourier electron density maps have been widely used in X-ray crystallography for half a century, they remain an essential tool for structural analysis. Moreover, the rapid pace at which the field of time-resolved serial X-ray crystallography is growing means that objective methods that improve the sensitivity of X-ray analysis to protein conformational changes have the potential to make a valuable contribution to the field. As the authors note, these tools may also help improve the interpretation of weakly binding compounds in structure-based drug design, which is another very important area of application. This article presents a well-argued, serious and novel approach to the problem of maximizing the confidence in difference Fourier electron density maps, and I recommend publication with only minor changes to the presentation of the article.

Specific comments on the article:

Abstract: if it is not too much of a diversion, a short definition of negentropy (reverse entropy?) may be appropriate in the abstract.

Introduction:

Page 4 introduces some of the widely used methods for calculating difference Fourier electron density maps. In my experience, the algorithm used by Phenix is often superior to some of the other tools, and it may be appropriate to comment on how the difference Fourier electron density map calculation is also implemented in Phenix.

Page 4: k-weighting is an important concept in this article. I therefore strongly recommend more background be given to this concept in the introduction, and perhaps an equation to define the approach may be appropriate. In general, although largely well written, the article sometimes assumes a lot of specialist expertise of the reader. By taking the time to introduce the key concepts more clearly, the authors should be able to take some of the less specialized readers with them.

Page 5: negentropy is defined in page 5, but it's a bit hanging as to how the reader should intuitively interpret this concept in this context. For example, it is stated that "negentropy is routinely applied as a measure of non-Gaussianity in independent component analysis", but this does little to convey the key idea to the non-specialist. How should the reader interpret this concept within the framework of using negentropy as a tool to improve the signal to noise in difference Fourier electron density maps?

Page 5: I wonder if TV is really a necessary acronym? It would not cost too much to use total variation throughout and certainly some of your readership will be old enough to remember when TV stood for a very different concept.

Page 6: I thought that the authors made a stronger attempt to communicate the concept of total variation denoising to the

reader, than for some of the other less familiar concepts in the article. Perhaps the authors could spare a few words to draw a clear distinction between total variation denoising and, say, a running average? Or have I got a wire crossed?

Page 6: I belong to the sub-set of X-ray crystallographers who understand that estimating phases for F' can, in principle, improve the signal to noise by a factor of two, but I remain suspicious of introducing model bias towards the unknown conformation into the difference density map calculation. On this occasion, however, I'll let the authors away with this since they do acknowledge the risks elsewhere in the paper.

Page 7: some strike-through edits remain in the manuscript on the top of page 7 and elsewhere in the article. The authors should be sure to correct this prior to resubmission.

Results:

Page 7: again I was looking for a definition of how negentropy is calculated in practice. Either an equation is used to define the value to be maximized, or a clear explanation of how this value is extracted. Likewise, the concepts of skewness and kurtosis remain undefined on Page 8. I think a bit more explanation of these key concepts would make the article more accessible to a structural biologist.

Page 8: Here it is stated that optimization of the commonly used k-weighting amplitude modification scheme is used when maximizing negentropy. Again, I think that an equation which explains the k-weighting scheme is appropriate, even though two appropriate citations are given in the text.

Page 9: Another concept, the power spectra of the difference electron density map, is discussed without a clear definition that takes the reader with the them. Please take the time to define this concept.

Page 10 and 11: the authors discuss the advantage of introducing phases for F' in the difference Fourier electron density map calculation. While the argument is correct, the risk of phase bias influencing the map is also important to acknowledge (I know this is acknowledged elsewhere, but it's such a crucial point it should always be clear).

Page 12: the improvement in terms of root mean square density of the map from +/- 5 rms to +/- 11 rms is impressive. I would advise that the improvement be checked independently. I am guilty in the past of quoting rms values from a script that incorporated a Bayesian weighting when calculating difference Fourier electron density maps, but it had a bug so that the numerical sigma values that could be read in COOT were stronger than was truthful. An improvement by almost a factor of two in rms is a very significant gain if correct.

Page 13: here it is commented that the method is able to recover chemically identifiable changes in electron density despite the phases of the excited conformation not being incorporated in the calculation (ie. an acknowledgement of the risks of phase bias).

Page 14: The authors add difference density to the resting state so as to recover a real-space model for model building. This is fine, but may be good to discuss how the approach differs from the implementation of extrapolated data in eg. Xtrapol8.

Page 16: again the k-parameter and lambda parameter make an appearance at the top of this page. As previously noted, I request that the k-parameter be defined in the article. Likewise, can the lambda parameter also be clearly defined.

Page 16: iterative refinement of total variation denoising is said to improve the maps. It may be useful to check what is iteratively being improved is being clearly communicated to the reader, since a structural biologist will be biased by the connection between the word "refinement" and "structural refinement of the coordinates". I don't think anything is incorrect in how it is expressed, but worth reading one more time to see if a point of potential confusion may be avoided.

Methods:

The description of how total variation denoising is implemented by the algorithm is rather technical. Is there any way to communicate to the non-specialist the key ideas?

Page 18-19: Perhaps a schematic illustrating how the phase estimates for the latent perturbed state is determined is worth including in the article? This seems to be given with the equations on page 19 (not numbered) and in Figure 3 panel a. In Figure 3, what is presumably the iteration number is not given on the x-axis. I am not completely certain what is happening when the phase is projected in Figure 3a (upwards vertical arrow). If the authors can improve the clarity of this schematic, it may be helpful. Looking at the supplementary material as I write this report, I see that this point is explained in Figure S6, so perhaps no changes are necessary.

Page 19: Why does the factor 2 emerge from the fact that $f = 2 \times \alpha$?

Page 21: I appreciate that the code for this tool has been released on github.

Figures:

Overall, the figures are very good and have been thoughtfully designed. But some things could be clearer.

Figure 1: What is meant by “Theoretical quantities” in the X-axis of the plots? I did not find this comparison with sample quantities easy to follow.

Figure 2: Again, the figure is largely clear, but perhaps the impact of the figure is compromised by trying to put too much information into the figure. I did not understand what is being plotted with the average power versus $d-1$.

Figure 3: See comment above.

Figure 4: There is a lot of information. Perhaps less is more, and the authors can reflect on whether or not the key message may be better communicated with fewer rather similar looking difference density maps.

Supporting information:

The supporting information contains a lot of figures to communicate several key ideas in this analysis, and to demonstrate how the advocated approach improves the interpretability of difference density maps. I do consider that there are a lot of details here that will assist the specialist as they work through the program on github, and try to implement this algorithm in their own work. On the other hand, it is rather detailed, and there are eleven supplementary figures. I would therefore recommend that the authors look at each and every supplementary figure and ask themselves how much additional value this adds to the main message of the article.

Final comment:

Overall, I intend that the authors use this feedback and their own judgement to improve the article. Please do not undertake a great deal of additional work if anything in my report is unclear. I support publication of the work, but if possible, I would like the main message of the article to be communicated better to the structural biology community.

Reviewer #2

(Remarks to the Author)

I read the work by Fadini et al. with much interest as the authors address long-standing challenges in the detection of conformational changes and ligand-binding events in a novel way that is a valuable addition to our collective toolkit. Their approach consists of three parts: a) introducing negentropy as a measure of difference map signal, b) introducing total variation denoising as a way to improve difference electron density maps, and c) introducing iterative phase improvement. Each of these aspects is innovative. In addition, they provide evidence that their difference maps may aid refinement of low-occupancy states relative to the state of the art for drug fragment detection, PanDDA.

By several measures, their proposed approach results in substantial improvements over current analytical approaches and/or successfully complements other approaches. I find the results in Figure S3 particularly convincing as they more clearly demonstrate improvement in map quality than some of the visual assessments that are both more subjective and dependent on choice of contour levels and display (see below; could Fig S3 be moved to the main text)? Figure 4 likewise shows real improvement in difference map quality (subject to some concerns expressed below).

The manuscript is written in a clear, accessible style and nicely illustrated, and should be of interest to the readership of Communications Biology. One could also wonder about applications to cryo-EM.

My main questions center on four key questions:

1. Are the observed increases in signal-to-noise ratio (sigma level) in the examples primarily due to the suppression of noise in the large swaths of space where no signal is likely to occur (decreasing sigma), rather than the improvement in actual signal? Would it be possible to adjust sigma levels to compensate for this effect? This could take the forms of looking at absolute e/A^3 levels side-by-side (like seems to happen in figure 2a for the 2D images), or, in the language of equation 1, can we directly compare the “s” and “u” maps? E.g. on line 239-240: “much stronger density” – is this true only in SNR units? For Fig 4, the authors could provide a version of Fig 4 in the SI that is more apples-to-apples from this point of view.

2. Is there a risk that the method invents signal (at least in sigma units), where there is none? “Negative controls” for the approach appear to be missing (e.g., synthetic or real dark – dark or apo – apo difference maps, or maps with scrambled differences).

3. Following the argument on lines 115-117 and line 125-126, to what extent does the method, conversely, “simply” act like a low-pass filter, suppressing high-frequency features in the difference map that are less likely to be true signal? This question has a few parts: (a) would a conventional low-pass filter (e.g., Gaussian) be able to achieve the same effect? (b) is there a case to be made that some type of low-pass filtering is indeed desirable and that the TV approach is better than others? I do not know the relevant literature in image analysis well, and neither will most readers. It would be helpful to bring us some insight; (c) to what extent are true sharp spatial features likely to be lost?

4. Although the software is, laudably, open-source, the authors do not provide access to their maps. Could they do so, e.g., through Zenodo or another repository?

Technical points:

1. Negentropy: it is not uncommon for structure factor amplitudes for different conditions to vary in average size after initial data processing. This has been addressed by various local scaling approaches implemented in, e.g., SOLVE, SCALEIT, and in estimation of the local Σ in implementations of the French-Wilson algorithm (e.g. in PHENIX). Without such local scaling, features of the density may contaminate the difference density. I could imagine such artifacts increasing negentropy but for the wrong reasons. It would be helpful for the authors to introduce a note of caution about the use of negentropy as a measure of difference map quality if they agree this concern is justified (referred to as such in lines 99, 106).

2. Outliers: Line 155-156: similar to comments above about imbalance between datasets, the invocation of the CLT suggests that meteor may not be robust to outlier ΔF 's and that k weighting, which suppresses outliers, may improve the reliability of meteor. Interestingly (lines 172-177), the authors use negentropy to tune k at least in one case. That seems risky if true outliers are present as I could imagine them increasing negentropy if not suppressed. Confusingly, line 429 also describes a k -weighting step during the iterative TV version of the algorithm without providing further detail. How? Is it important?

Minor points

The authors may want to refer to METEOR in their title or abstract. This would not be the first crystallographic software for which the corresponding paper would be hard to find.

Notably, the authors refer to $<30\%$ occupancy as low (line 75) and then apply their method in Figure 4 and lines 252-256 to ligands of higher occupancy. It would be more interesting to see the effect on difference maps of lower-occupancy ligands.

Line 88: "this manual intervention requires extensive trial-and-error ..." – to my limited knowledge, such optimization is not commonplace, but I could be wrong.

There are several instances of residual text editing (lines 140, 172, 221, 319, ...)

Line 142: "are compatible and stackable with existing methods" – is there any potential for complications the reader should worry about? Are there any particularly interesting cases of stacking where meteor may complement another approach well?

Figure S1: The description of n as the "noise fraction" is highly confusing. Reading the supplement, it becomes clear that n is the fraction of pixels in the true difference density map with negligible difference density, rather than something like $1 -$ (excited state fraction). Going by the examples in Fig 1a-c, I would doubt whether any scenario with $n < 0.99$ or so is ever relevant in practice!

The example in Figure S1 feels rather contrived. Perhaps the authors could take the histograms in Figure 1a-c, all of which have a dominant central mode in their noise-free DED histogram, and add synthetic noise as in Fig S1. I would be surprised if, in such an exercise, the kurtosis would exhibit non-monotonic dependence on n (or more importantly, σ_n !).

Line 211: "additive noise" --- added in real space or reciprocal space?

Figure 2b, power spectrum: it would be nice to have a log-spaced x-axis like in 2a as most of the interesting information, at high resolution, is essentially uninterpretable in the current display. It is also hard to tell apart the colors in these power spectra.

Figure 3: What are the units of phase change? Confusingly, the caption refers to 3c showing the mean (absolute?) change in phase, while the main text (line 237) speaks of cumulative phase change—which, I would assume, goes up rather than down. At the same time, if this is the phase change per iteration, the cumulative change would seem very large if the units are radians rather than degrees! Also welcome would be a measure of deviation from an Fc-Fc map (real-space correlation or, e.g., $\langle \cos(\Delta\phi) \rangle$).

Line 364: "... potentially doubling ..." – I can see why the authors might say this but it seems like an exaggeration at present.

Line 370-371: is this a relevant concern? If so, when?

Line 376-377: what is the statement based on that Fo-Fo maps "contain weaker signals" than "Fo-Fc" maps (are modeling errors signals?)?

Lines 419-429: It would be helpful to guide the reader more explicitly through Figure S6. This is a key figure but hard to digest.

Line 480: pandas, not panda?

Lines 629-633: "from a standard normal distribution" – of the same variance?

Figure S8b: the legend seems to cover the interesting part of the graph.

Reviewer #3

(Remarks to the Author)

The manuscript "Denoising Reveals Low-Occupancy Populations in Protein Crystals" by Professor van Thor and colleagues [Reference COMMSBIO-24-8699-T], presents an innovative framework that integrates negentropy and total variation (TV) denoising to enhance difference electron density (DED) maps, addressing key challenges in low-occupancy signal detection for structural biology. This methodology represents a significant advancement in the field by introducing negentropy as a robust metric to optimize difference map parameters, offering an automated and reproducible alternative to traditional trial-and-error approaches. Furthermore, the iterative-TV (it-TV) approach is a notable development, allowing for iterative phase refinement to improve the interpretability of weak signals. These innovations are complemented by validation using both synthetic and experimental datasets, demonstrating the broad applicability of the method in diverse contexts such as time-resolved crystallography and fragment screening.

The authors skillfully justify their choice of negentropy as a measure of map quality, showing that it effectively quantifies non-Gaussianity in voxel value distributions and is superior to other metrics like skewness and kurtosis. The integration of TV denoising significantly improves the signal-to-noise ratio in DED maps, as demonstrated in both single-pass and iterative applications. Importantly, the manuscript highlights the practical advantages of their approach, including its accessibility as an open-source package (METEOR) and its computational efficiency, making it suitable for real-time experimental workflows and high-throughput campaigns.

The authors' innovative use of negentropy and TV denoising addresses longstanding challenges in the analysis of low-occupancy states, offering a broadly applicable solution that is particularly relevant for time-resolved and fragment-screening crystallography. The figures effectively illustrate the method's capabilities, with clear improvements in signal clarity and interpretability across all test cases. For example, the application of it-TV to datasets involving CI-rsEGFP2 photoisomerization and fragment-bound SARS-CoV-2 main protease provides striking evidence of the method's ability to reveal chemically meaningful features that were previously obscured by noise.

However, there are some areas that would benefit from further discussion.

1) While the authors acknowledge the limitations of the it-TV approach in high-noise conditions due to potential susceptibility to local minima, they could provide a more detailed discussion of how these challenges might be mitigated in future developments. Additionally, the manuscript primarily addresses Gaussian noise and does not explicitly consider the effects of systematic errors common in crystallographic datasets, such as scaling artifacts or anisotropy. While these limitations do not detract from the immediate utility of the method, acknowledging them more explicitly and suggesting potential solutions or complementary approaches would strengthen the manuscript.

2) Another point of consideration is computational scalability. While the authors describe their approach as computationally efficient, more specific information about runtime performance across datasets of varying sizes would provide valuable context, particularly for users planning to apply this method in high-throughput scenarios.

3) Furthermore, while the manuscript effectively demonstrates the method's strengths relative to existing tools like PanDDA and Xtrapol8, a more in-depth qualitative comparison with emerging deep learning-based denoising techniques would help contextualize the advantages of TV denoising.

Despite these considerations, the manuscript is a strong candidate for publication in *Communications Biology*. Its methodological innovations are timely and impactful, addressing critical needs in the field. While additional benchmarking or testing under non-Gaussian noise conditions could further enhance the work, these aspects can be reasonably addressed through expanded discussion rather than new experiments. Overall, this study advances the state of the art in DED map analysis and is likely to have a lasting impact on both methodological development and practical applications in structural biology.

Version 1:

Reviewer comments:

Reviewer #1

(Remarks to the Author)

All three reviewer reports from the previous version of this manuscript were constructive and supporting publication. The authors have made serious and constructive efforts to modify the article according to the suggestions of reviewers, and the article is more accessible as a consequence. In particular, the authors have been very constructive in their response to the comments from my initial report. I therefore recommend publication without any further changes.

I apologize both to the authors and the editorial team for my delay in providing this report. The request came when I was on vacation, and my return to work was a sequence of deadlines which meant that I could only turn attention to this article today.

Reviewer #2

(Remarks to the Author)

The author have carefully and comprehensively addressed my questions. A few small remaining things:

p. 9 of the rebuttal, regarding use of the term "refinement". It sounds like the authors were about to replace the phrase "iterative refinement" with something else but then didn't? I am fine with the current wording but wonder if it is indeed the intended rewording.

Figure 3C: the x-axis label is still missing.

Reviewer #3

(Remarks to the Author)

The authors have satisfactorily addressed all of my previous comments and concerns in the revised manuscript. I find the current version to be scientifically sound and clearly presented. I therefore support its publication in Communications Biology.

RESPONSE TO REVIEWERS

REVIEWER 1

This article describes an approach to improving the signal-to-noise ratio in difference Fourier electron density maps so that weak difference signals may be resolved with improved confidence. Although difference Fourier electron density maps have been widely used in X-ray crystallography for half a century, they remain an essential tool for structural analysis. Moreover, the rapid pace at which the field of time-resolved serial X-ray crystallography is growing means that objective methods that improve the sensitivity of X-ray analysis to protein conformational changes have the potential to make a valuable contribution to the field. As the authors note, these tools may also help improve the interpretation of weakly binding compounds in structure-based drug design, which is another very important area of application. This article presents a well-argued, serious and novel approach to the problem of maximizing the confidence in difference Fourier electron density maps, and I recommend publication with only minor changes to the presentation of the article.

We would like to thank the reviewer for their thoughtful and encouraging comments – we are glad the significance and potential impact of our approach came through clearly.

Specific comments on the article:

Abstract: if it is not too much of a diversion, a short definition of negentropy (reverse entropy?) may be appropriate in the abstract.

We added an intuitive description of negentropy in the following sentence of the abstract:

We address these issues, first by showing that negentropy – a measure of how different a signal looks from anticipated Gaussian noise – is an effective metric to assess difference map quality and can therefore be used to automatically determine parameters needed during difference map calculation.

We hope this addresses the reviewer's point while also maintaining the scope of a broad readership.

Introduction:

Page 4 introduces some of the widely used methods for calculating difference Fourier electron density maps. In my experience, the algorithm used by Phenix is often superior to some of the other tools, and it may be appropriate to comment on how the difference Fourier electron density map calculation is also implemented in Phenix.

Good point. We now introduce Phenix on page 4:

the Phenix suite computes isomorphous difference maps by normalizing the common set of reflections for F'_{obs} and F_{obs} and scaling these to each other using either a single scale factor or a multi-scale protocol.

Page 4: k-weighting is an important concept in this article. I therefore strongly recommend more background be given to this concept in the introduction, and perhaps an equation to define the approach may be appropriate. In general, although largely well written, the article sometimes assumes a lot of specialist expertise of the reader. By taking the time to introduce the key concepts more clearly, the authors should be able to take some of the less specialized readers with them.

This is also a good point. We thought introducing it with all the other methods on page 4 would be a good place:

One of the key weighting strategies implemented in Xtrapol8 is k-weighting, which adjusts the amplitude of each difference structure factor to reduce the influence of outliers. The weight applied to each difference structure factor amplitude (ΔF) is calculated as:

$$w = [1 + (\sigma^2_{\Delta F} / \langle \sigma^2_{\Delta F} \rangle) + k \cdot (|\Delta F|^2 / \langle |\Delta F|^2 \rangle)]^{-1}$$

where $\sigma_{\Delta F}$ is the uncertainty associated with a specific ΔF , and k is a tunable scaling factor to reduce the influence of outlier $|\Delta F|$ values with underestimated uncertainties. The magnitude of k controls how strongly large values of ΔF are suppressed. Implementations using $k=1$, $k=0$ and intermediate values of k have all been applied in the literature.

Page 5: negentropy is defined in page 5, but it's a bit hanging as to how the reader should intuitively interpret this concept in this context. For example, it is stated that "negentropy is routinely applied as a measure of non-Gaussianity in independent component analysis", but this does little to convey the key idea to the non-specialist. How should the reader interpret this concept within the framework of using negentropy as a tool to improve the signal to noise in difference Fourier electron density maps?

To intuitively introduce negentropy, we've added the following on page 6:

Map negentropy ($J(p_\rho)$) provides a useful metric for assessing how much structured, signal-like information is present in a difference map: a higher negentropy suggests the presence of interpretable features such as real electron density changes, rather than noise. Formally, it is the difference in differential entropy between the distribution of DED map voxel values (p_ρ) and a Gaussian distribution

with the same variance (p_{gauss}) (under the assumption that individual voxels are identically and independently distributed) [new citations added: Brillouin, 1953 (25); Schrödinger, 1944 (26)]:

$$J(\rho_\rho) = H(\rho_{\text{gauss}}) - H(\rho_\rho)$$

where the differential entropy $H(\rho_\rho)$ is defined as:

$$H(\rho_\rho) = -\int \rho_\rho(u) \cdot \log \rho_\rho(u) du$$

Summed over all voxels (u). Already routinely applied in independent component analysis (ICA) as a measure of non-Gaussianity, negentropy quantifies how far a signal deviates from Gaussian randomness. We show that maximizing negentropy is an effective approach for selecting parameters in models that aim to denoise DED maps.

Page 5: I wonder if TV is really a necessary acronym? It would not cost too much to use total variation throughout and certainly some of your readership will be old enough to remember when TV stood for a very different concept.

We appreciate the reviewer's comment and recognize the potential for ambiguity in the acronym TV. However, we have chosen to retain it, as TV (for *total variation*) is a well-established and widely used abbreviation in the image processing and mathematical optimization literature, and it also benefits from being a concise, recognizable term. We think its conciseness makes it helpful for readability, particularly given the frequency with which the method appears in the manuscript.

Page 6: I thought that the authors made a stronger attempt to communicate the concept of total variation denoising to the reader, than for some of the other less familiar concepts in the article. Perhaps the authors could spare a few words to draw a clear distinction between total variation denoising and, say, a running average? Or have I got a wire crossed?

We thank the reviewer for this suggestion and note that it also relates to a comment by Reviewer 2. To clarify the nature of total variation denoising, we have added a sentence explicitly contrasting it with other denoising filters:

Page 7: TV denoising differs from commonly used low-pass filters, such as running averages or Gaussian filtering, by preserving sharp features (such as edges or peaks) while selectively suppressing small-scale fluctuations characteristic of noise. In contrast, low pass filters suppress high-frequency noise but blur sharp signal features. TV denoising is therefore particularly well-suited to cases where enhancing sparse, localized signals is important, making it a promising approach for DED maps.

Page 6: I belong to the sub-set of X-ray crystallographers who understand that estimating phases for F' can, in principle, improve the signal to noise by a factor of two, but I remain suspicious of

introducing model bias towards the unknown conformation into the difference density map calculation. On this occasion, however, I'll let the authors away with this since they do acknowledge the risks elsewhere in the paper.

This is definitely an important point. We fully agree that introducing phase estimates can carry the risk of model bias. This is why we emphasize later in the manuscript that our approach does not introduce or rely on an atomic model for the unknown conformation.

See below for our response, including new text added to the manuscript explicitly discussing phase bias.

Page 7: some strike-through edits remain in the manuscript on the top of page 7 and elsewhere in the article. The authors should be sure to correct this prior to resubmission.

Fixed. We thank the reviewer for catching this!

Results:

Page 7: again I was looking for a definition of how negentropy is calculated in practice. Either an equation is used to define the value to be maximized, or a clear explanation of how this value is extracted. Likewise, the concepts of skewness and kurtosis remain undefined on Page 8. I think a bit more explanation of these key concepts would make the article more accessible to a structural biologist.

Thank you for pointing this out.

We introduced a more detailed definition of negentropy in the context of DED maps in the introduction on page 6, see our reply above to another comment from the reviewer.

Further, we also now include brief definitions of skewness and kurtosis on page 9 to improve accessibility for structural biology readers who may be less familiar with these statistical measures:

Skewness, kurtosis, and negentropy are well-known measures for non-Gaussianity. In addition to negentropy, described above, skewness measures asymmetry in a distribution, while kurtosis describes the "tailedness" or sharpness of its peak. To investigate the suitability of these statistics as indicators of difference map quality (...)

Page 8: Here it is stated that optimization of the commonly used k-weighting amplitude modification scheme is used when maximizing negentropy. Again, I think that an equation which explains the k-weighting scheme is appropriate, even though two appropriate citations are given in the text.

Indeed – we agreed with the first comment the reviewer had on k-weighting and have included the equation and definition on page 4 (see response above).

Page 9: Another concept, the power spectra of the difference electron density map, is discussed without a clear definition that takes the reader with them. Please take the time to define this concept.

We have added a brief explanation of what a power spectrum represents in the context of difference electron density maps on page 11:

To support this subjective assessment, we plot the power spectra of the DED maps. The power spectrum of a difference map describes how the signal is distributed across spatial frequencies (resolution shells), with higher frequencies corresponding to finer structural detail. By comparing power spectra before and after denoising, we can assess whether the process has effectively suppressed high-frequency noise while preserving meaningful signal.

Page 10 and 11: the authors discuss the advantage of introducing phases for F' in the difference Fourier electron density map calculation. While the argument is correct, the risk of phase bias influencing the map is also important to acknowledge (I know this is acknowledged elsewhere, but it's such a crucial point it should always be clear).

We agree with the reviewer that acknowledging the risk of phase bias explicitly and more extensively is a good idea. We have added the following paragraph to the discussion on page 19:

Like solvent flattening and phase retrieval, the it-TV method does not involve an atomic model or user intervention and thus avoids model-driven phase bias. That said, phase modification could, in principle, introduce spurious features that might be biologically interpreted. To counter this possibility, we recommend comparing it-TV maps with both single-pass TV maps and unprocessed maps. When denoising is successful, these maps should be consistent and not in contradiction.

Page 12: the improvement in terms of root mean square density of the map from +/- 5 rms to +/- 11 rms is impressive. I would advise that the improvement be checked independently. I am guilty in the past of quoting rms values from a script that incorporated a Bayesian weighting when calculating difference Fourier electron density maps, but it had a bug so that the numerical sigma values that could be read in COOT were stronger than was truthful. An improvement by almost a factor of two in rms is a very significant gain if correct.

This is a legitimate concern so we carried out an additional analysis to confirm the signal strengths. To ensure that COOT's contour levels were correct, we analyzed DED map voxel value histograms

before and after denoising. Below are histograms for the CI-rsEGFP2 example used in the main text (y-axis capped at 500 below to focus on voxels with large values):

As already discussed in depth in the manuscript, the original DED map prior to any denoising (left) is essentially Gaussian. The probability of finding values above/below 6 rms for this map is thus extremely low – only 0.0000002% of values fall outside this range for a Gaussian – and this is reflected by the first histogram. On the other hand, we can see that there are signals above/below +/- 6 rms for the TV denoised map (histogram on the right).

Below are the RMS statistics for the original and TV denoised DED maps:

Original map

RMS: 0.020

Minimum value: -0.093 (-4.66 × RMS)

Maximum value: 0.090 (4.51 × RMS)

TV denoised map

RMS: 0.015

Minimum value: -0.210 (-13.58 × RMS)

Maximum value: 0.201 (13.04 × RMS)

To confirm the high rms values correspond to the signals we expect on the chromophore, we loaded the rescaled TV denoised map in Coot at +/- 6 rms:

As an additional test, we kept map values for voxels at +/- 6 rms in the map and set all voxels for the intermediate range (-6 rms < voxel value < 6 rms) to zero. The resulting maps, in which most voxel values have been set to zero, are overlaid in black below:

This procedure ends up setting *all* the voxels in the non-denoised map to zero. For the TV-denoised map, when we load this in COOT we can indeed confirm that those high rms voxels report on the chromophore signals (the rest of the map is empty):

This gives us confidence in our stated rms countour values.

We considered adding these tests to the supplement, but since they are not necessary to understand our results and the reviewer recommended that we limit and focus the supplementary information as much as possible, we elected to not include this analysis in our revised manuscript.

Page 13: here it is commented that the method is able to recover chemically identifiable changes in electron density despite the phases of the excited conformation not being incorporated in the calculation (ie. an acknowledgement of the risks of phase bias).

Definitely – see our expansion of the discussion around phase bias in the response above.

Page 14: The authors add difference density to the resting state so as to recover a real-space model for model building. This is fine, but may be good to discuss how the approach differs from the implementation of extrapolated data in e.g. Xtrapol8.

Good point. Our goal for this section of the results is to exemplify how TV-denoised maps can be used to support model building by improving the quality and interpretability of extrapolated density, but not address extrapolation strategies *per se*. There are well-established strategies for extrapolation, both in real (PanDDA) and reciprocal space (Xtrapol8), and we don't add anything specifically new in that regard in our manuscript. Therefore, we simply chose to illustrate the real-space approach as a straightforward example.

To clarify this, we changed the text on page 16 to avoid giving the impression that we are introducing a new extrapolation framework:

To produce an extrapolated map, an accurate estimate of the perturbed state occupancy is essential: once an estimate for the occupancy is known, the extrapolated map can be computed by performing

an appropriate addition in real space (between a reference map and a difference map, as done by PanDDA) or in reciprocal space (between reference structure factors and difference structure factors, as done by Xtrapol8). We use the real-space approach here to illustrate how TV denoised difference maps can improve interpretability.

Page 16: again the k-parameter and lambda parameter make an appearance at the top of this page. As previously noted, I request that the k-parameter be defined in the article. Likewise, can the lambda parameter also be clearly defined.

This reviewer made a similar comment requesting elaboration of k-weighting, which we have now provided (see our response and the changes to the manuscript above).

For the lambda regularization parameter used in total variation denoising, a full explanation and the underlying equation are provided in Supporting Note S1 of the Supplementary Information. To ensure this is easier to locate, we have now added a pointer to the SI and a brief description in the main text where lambda is first introduced (see page 10):

A necessary step in TV denoising is to choose the degree of smoothing, dictated by a regularization parameter, λ , which controls the trade-off between denoising strength and how closely the result matches the original map (see Supporting Note S1 for definition and formulation)

Page 16: iterative refinement of total variation denoising is said to improve the maps. It may be useful to check what is iteratively being improved is being clearly communicated to the reader, since a structural biologist will be biased by the connection between the word “refinement” and “structural refinement of the coordinates”. I don’t think anything is incorrect in how it is expressed, but worth reading one more time to see if a point of potential confusion may be avoided.

This is a very good point. While the use of “iterative refinement” is technically accurate in the context of improving map phases, we agree that it may unintentionally suggest coordinate refinement (especially to structural biologists) and it’s best to avoid this possible confusion. We have revised the wording on page 18:

Across all three test cases, the it-TV output maps exhibit greater chemical interpretability and higher negentropy than those produced by a single pass of TV denoising, suggesting that iterative refinement adds significant value.

Methods:

The description of how total variation denoising is implemented by the algorithm is rather technical. Is there any way to communicate to the non-specialist the key ideas?

We appreciate this comment and agree that the technical description of the algorithm may not interest all readers. For this reason, we have restricted the technical discussion to the SI and we've aimed to make the key ideas more accessible through the addition of intuitive explanations in the Results section (page 7):

The total variation is simply defined as the sum of the changes from each voxel to all neighboring voxels; minimizing this quantity suppresses small-scale noise while preserving important structural features. Applied as a density modification technique, TV denoising thus produces a map that closely resembles the input map, but with flattened noise-dominated regions and preserved peaks or edges. TV denoising differs from commonly used low-pass filters, such as running averages or Gaussian filtering, by preserving sharp features (such as edges or peaks) while selectively suppressing small-scale fluctuations characteristic of noise. In contrast, low pass filters suppress high-frequency noise but blur sharp signal features. TV denoising is therefore particularly well-suited to cases where enhancing sparse, localized signals is important, making it a promising approach for DED maps.

In addition, we've visualized effect of different lambda values in Figure 2.

We hope this helps non-specialists grasp the core concept of how total variation denoising sharpens signal while suppressing noise.

Page 18-19: Perhaps a schematic illustrating how the phase estimates for the latent perturbed state is determined is worth including in the article? This seems to be given with the equations on page 19 (not numbered) and in Figure 3 panel a. In Figure 3, what is presumably the iteration number is not given on the x-axis. I am not completely certain what is happening when the phase is projected in Figure 3a (upwards vertical arrow). If the authors can improve the clarity of this schematic, it may be helpful. Looking at the supplementary material as I write this report, I see that this point is explained in Figure S6, so perhaps no changes are necessary.

Indeed, the process of estimating latent perturbed-state phases is illustrated in more detail and more intuitively in Figure S6, where we include an Argand diagram to clarify the projection step referenced in Figure 3a.

However, the reviewer correctly pointed out that our x-axis label (iteration number) in Figure 3c was missing, so we have corrected that! We have also improved the legend for Figure S6 to better help the reader (Reviewer 2, below, highlighted this issue as well).

Page 19: Why does the factor 2 emerge from the fact that $f = 2 \times \alpha$?

The factor of 2 in the relationship $f=2\alpha$ arises from assumptions made under the constant phase approximation and is derived in detail in the literature. For brevity and clarity, we felt it was best not

to reproduce the full mathematical treatment in the main text, but we have added the relevant citations after the statement (Ref. 37 in the manuscript).

Page 21: I appreciate that the code for this tool has been released on GitHub.

We are very glad to make the code available and hope it will be useful to the community. We welcome feedback from users and hope to improve the tool further based on their applications!

Figures:

Overall, the figures are very good and have been thoughtfully designed. But some things could be clearer.

Figure 1: What is meant by “Theoretical quantities” in the X-axis of the plots? I did not find this comparison with sample quantities easy to follow.

In our plots, *theoretical quantiles* refer to values that would be expected if the data followed a perfect Gaussian (normal) distribution. The *sample quantiles*, in contrast, are the actual values observed in our data. We use this comparison to visually assess how well the map distributions match the assumptions of a Gaussian distribution. This visualization of probability plots and quantiles is common practice in statistics, however, the reviewer’s comment points out that we may need to be more clear, especially for readers without a statistical background. We have revised the figure legend to use more intuitive language:

Fig 1 legend: (...) On the right column, the normal probability plot compares the observed map voxel value distribution to what would be expected if the data followed a perfect Gaussian (normal) distribution: the observed data (sample quantiles) are ordered and plotted against the expected values of the ordered statistics for a sample from a standard normal distribution (mean 0, variance 1) of the same size as the data (theoretical quantiles). If the observed values closely follow the red diagonal line, this indicates that the data are approximately Gaussian. Deviations from the line suggest departures from normality (...)

Figure 2: Again, the figure is largely clear, but perhaps the impact of the figure is compromised by trying to put too much information into the figure. I did not understand what is being plotted with the average power versus $d-1$.

Thank you for the feedback. We have edited the axis labels and plot coloring of the power spectra in Fig 2 to aid clarity. Further, we added the following text to page 11 introducing the concept of the power spectrum:

The power spectrum of a difference map describes how the signal is distributed across spatial frequencies (resolution shells), with higher frequencies corresponding to finer structural detail. By comparing power spectra before and after denoising, we can assess whether the process has effectively suppressed high-frequency noise while preserving meaningful signal.

Figure 3: See comment above.

Addressed above.

Figure 4: There is a lot of information. Perhaps less is more, and the authors can reflect on whether or not the key message may be better communicated with fewer rather similar looking difference density maps.

We received contrasting feedback from reviewers in terms of level of detail to have in the main text Figure 4: while Reviewer 1 encouraged reducing the information shown in the main text figure, Reviewer 2 suggested elevating content such as Figure S3 into the main text, which would introduce even more detail to the figure. We discussed this and, to balance the perspectives, we have retained Figure S3 in the Supplementary Information but have revised the legend for main Figure 4 to explicitly point to the data and non-Gaussianity plots shown in Figure S3, directing the reader wanting more quantitative information.

Supporting information:

The supporting information contains a lot of figures to communicate several key ideas in this analysis, and to demonstrate how the advocated approach improves the interpretability of difference density maps. I do consider that there are a lot of details here that will assist the specialist as they work through the program on github, and try to implement this algorithm in their own work. On the other hand, it is rather detailed, and there are eleven supplementary figures. I would therefore recommend that the authors look at each and every supplementary figure and ask themselves how much additional value this adds to the main message of the article.

We appreciate the reviewer's comment. We've received differing feedback across the reviews – with Reviewers 2 and 3 encouraging the inclusion of additional detail, and Reviewer 1 expressing concern about too much complexity. In response, we've taken a middle-ground approach: we have reviewed each supplementary figure and believe each is helpful in supporting key technical points or guiding implementation, addressing a level of detail requested by the other two reviewers. We hope their inclusion in the SI rather than in the main text strikes a reasonable balance between accessibility and reproducibility.

Overall, I intend that the authors use this feedback and their own judgement to improve the article. Please do not undertake a great deal of additional work if anything in my report is unclear. I support

publication of the work, but if possible, I would like the main message of the article to be communicated better to the structural biology community.

We sincerely thank the reviewer for their constructive feedback throughout. We have made several changes in direct response to the points raised, with a particular focus on improving the accessibility and clarity of the manuscript for the broader structural biology community.

REVIEWER 2

I read the work by Fadini et al. with much interest as the authors address long-standing challenges in the detection of conformational changes and ligand-binding events in a novel way that is a valuable addition to our collective toolkit. Their approach consists of three parts: a) introducing negentropy as a measure of difference map signal, b) introducing total variation denoising as a way to improve difference electron density maps, and c) introducing iterative phase improvement. Each of these aspects is innovative. In addition, they provide evidence that their difference maps may aid refinement of low-occupancy states relative to the state of the art for drug fragment detection, PanDDA.

By several measures, their proposed approach results in substantial improvements over current analytical approaches and/or successfully complements other approaches. I find the results in Figure S3 particularly convincing as they more clearly demonstrate improvement in map quality than some of the visual assessments that are both more subjective and dependent on choice of contour levels and display (see below; could Fig S3 be moved to the main text)? Figure 4 likewise shows real improvement in difference map quality (subject to some concerns expressed below).

The manuscript is written in a clear, accessible style and nicely illustrated, and should be of interest to the readership of Communications Biology. One could also wonder about applications to cryo-EM.

We thank the reviewer for their thoughtful assessment of our work. We are particularly encouraged by their recognition of the innovations and the practical gains demonstrated in our examples and we hope our responses below now address the questions raised.

We discussed the specific mention of Figures S3 and Figure 4 in supporting our claims given that we received contrasting feedback from different reviewers in terms of level of detail to have in the main text: while Reviewer 1 encouraged reducing the information shown in the main text figures (particularly Figure 4), this reviewer suggested elevating content such as Figure S3 into the main text. To balance these perspectives, we have retained Figure S3 in the Supplementary Information but have revised the legend for main Figure 4 to explicitly point to the data and non-Gaussianity plots shown in Figure S4 and S5.

Addition to the end of Figure 4 legend: *The improved interpretability seen here is further supported by the quantitative distribution analysis shown in Supplementary Figures S4-S5.*

In response to the reviewer's suggestion regarding potential applications to cryo-EM, we have added a sentence to the Conclusions section to acknowledge this exciting direction for future development.

Page 22: *Beyond crystallography, we anticipate that this general framework could prove useful in related structural techniques such as cryo-EM, where weak, low-occupancy signals in maps are also a limiting factor due to partial ligand binding or conformational variability.*

My main questions center on four key questions:

1. Are the observed increases in signal-to-noise ratio (sigma level) in the examples primarily due to the suppression of noise in the large swaths of space where no signal is likely to occur (decreasing sigma), rather than the improvement in actual signal? Would it be possible to adjust sigma levels to compensate for this effect? This could take the forms of looking at absolute $e/\text{\AA}^3$ levels side-by-side (like seems to happen in figure 2a for the 2D images), or, in the language of equation 1, can we directly compare the “s” and “u” maps?

In response to the first question here, simply put, yes – the observed improvements in signal-to-noise ratio (SNR) are primarily driven by the suppression of noise in regions of the map with no difference electron density signal. As with all denoising techniques, TV denoising improves SNR by preferentially attenuating noise power significantly more than any attenuation of the power of the signal.

The reviewer's comment led us to the important realization that application of a TV denoising procedure (or in fact of any amplitude modification technique, including k-weighting) changes the scale of the map. The electron density values of the input map have arbitrary units (an unknown scalar multiplied by $e/\text{\AA}^3$), but the output map is on a different arbitrary scale. When one visualizes these maps by contouring a specific rms value, these arbitrary units divide out, so there is no substantial difference in, for example, Coot visualization. However, we found an arbitrary change of scale undesirable, and sought to ensure that the input and output maps would in fact have the same units. If we analyze the power spectra of the maps before or after denoising, we indeed observe that the total power of the map is decreased by TV filtering. Below we include the plot for the Cl-rsEGFP2 data used throughout the main text (“original” refers to a vanilla, non-denoised Fo-Fo map):

Note, as mentioned, that the reduction in power is not just a feature of TV denoising, but also of k-weighting: this has not been commented on in the literature, despite the wide adoption of k-weighting by the time-resolved community.

The reviewer’s second question here is therefore very relevant and made us realize we could try to adjust for a reduction in power whenever the original set of ΔF s is modified. As the reviewer suggests, one can rescale the denoised map (u) back to the scale of the original map (s).

This concept is analogous to **gain normalization** in signal processing, where a denoised signal is rescaled to match the expected power of the original input *in signal-dominated frequency bands*. For example, in audio signal processing, it is common practice to normalize audio signal after denoising to adjust the amplitude of the cleaned signal to a desired level (this ensures consistency in loudness and prevents issues like clipping or distortion). In the case of difference maps, we reasoned that the signal-rich region of the map that will be least affected by the TV filtering process is the low-resolution regime. We can therefore scale the denoised map back to the original map using low resolution reflections. In METEOR we now use reflections that are lower than 1.5 \AA from the stated resolution cutoff (below 0.1 \AA^{-2} for the CI-rsEGP2 dataset).

Having applied this normalization, we can analyze the map voxel values before and after denoising with a clearer correspondence:

We can now compare the original map and the TV denoised + rescaled map, as we have done in our response to Reviewer 1 (please refer to response above):

to show that the regions of signal on the chromophore are indeed stronger in absolute terms.

The reviewer's comments helped us realize that this is an important point and deserves clearer emphasis in the manuscript. The application of a general scale factor to the final map does not affect our statements about rms values or the nature of the map voxel value distributions across the manuscript. It is, however, important to account for the difference in overall power when using the modified set of deltaFs (k-weighted, denoised, or both) for occupancy extrapolation, so we have

added the rescaling to the original map as a final step after TV. We now describe this process (and its reasons) in the Methods (page 22):

We have observed that schemes which modify difference structure factor amplitudes affect the total power of the related DED map. For example, both k -weighting and subsequent TV denoising result in a reduction in power compared to a simple Fo-Fo map. We show this is true for the Cl-rsEGFP2 data used throughout our work in Figure S13. It is important to account for this change in overall power when using a modified set of difference structure factors (k -weighted, denoised, or both) for occupancy extrapolation. To ensure that METEOR maps are on the same scale as the original inputs, we rescale weighted or denoised maps back to the corresponding original difference map. To do this, we apply a strategy analogous to gain normalization in signal processing, where a denoised signal is rescaled to match the expected power of the original input in signal-dominated frequency bands. For example, in audio signal processing, it is common practice to normalize audio signal after denoising to adjust the amplitude of the cleaned signal to a desired level (this ensures consistency in loudness and prevents issues like clipping or distortion). In the case of difference maps, we reasoned that the signal-rich region of the map that will be least affected by the TV filtering process is the low-resolution regime. We therefore scale METEOR maps back to original input maps using low resolution reflections (Figure S13). The default is to use reflections that are lower than 1.5 \AA from the stated resolution cutoff. We additionally include a comparison between the application of TV denoising and two other common types of filtering procedures to DED maps in Figure S14.

and include the power spectra above in the SI.

E.g. on line 239-240: “much stronger density” – is this true only in SNR units? For Fig 4, the authors could provide a version of Fig 4 in the SI that is more apples-to-apples from this point of view.

We agree the wording here was not precise and have replaced “*much stronger density*” with “*notably higher signal-to-noise difference signal*”. Our analysis above addresses the more general concern.

2. Is there a risk that the method invents signal (at least in sigma units), where there is none? “Negative controls” for the approach appear to be missing (e.g., synthetic or real dark – dark or apo – apo difference maps, or maps with scrambled differences).

This is a sharp comment and we were indeed missing this essential control experiment. Since the preprint release, we have in fact observed that when METEOR is applied to cases where there is no appreciable signal to denoise, a map with meaningless features is produced – but that the resulting “noise” features differ in nature from the Gaussian noise that users (including us!) might expect.

We built intuition by denoising simulated maps with synthetic noise as well as experimental maps absent of meaningful signal, including two real-life examples from users who contacted us.

In these cases, METEOR's behavior is consistent with the well-documented case of TV denoising of pure noise: studies have identified a "staircasing" effect (new reference added to main text on page 20, reference 46), which produces an output containing sharp edges in regions that should be "flat". This is expected, as TV denoising aims to flatten locally connected regions, and will therefore flatten and combine regions of noise that randomly happen to be similar in amplitude if this is the only "signal" present.

We aimed to alert users as to when it occurs and try to warn them proactively that they may be denoising noise (or signal/noise ratios that are too small even for METEOR). We have identified two useful indicators of such failure cases:

1. **The optimized regularization parameter λ becomes unusually large** (> 0.1 in our experience), which suggests that the algorithm is applying strong smoothing in the absence of recoverable signal.
2. **The resulting features are not spatially correlated with any known atoms** or structural differences in the model and often appear in unoccupied solvent regions.

To address this risk, we have now added a warning output to METEOR if λ exceeds 0.1 after optimization and included a note of caution in the revised manuscript (page 19-20) and online documentation, explicitly recommending that users visually inspect the resulting features and compare to the reference structure: TV denoising will not reveal signals that were previously unobservable, rather, by suppressing noise elsewhere, surface signals that may have been buried by noise (i.e. that were previously only visible at low rms).

Below is the synthetic example suggested by the reviewer. We computed the empty apo-apo map from the 7AWR/7AR6 datasets and added Gaussian noise (mean 0 and standard deviation 1) to the initially empty array. While the output map is not completely empty, our internal control catches the problematic behavior:

METEOR output:

```
[warning ] TV regularization weight much larger than expected, something probably went wrong limit=0.1 weight=0.24
```

We have incorporated the following paragraph in our Discussion:

Page 20: A further note of caution relates to the behavior of TV denoising in the absence of signal. In such cases TV minimization may still produce localized, structured features due to the well-known “staircasing” effect [new reference, 46]. This occurs because the method encourages piecewise-smooth output and may interpret random fluctuations as edges when no real signal is present. To help identify such false positives, METEOR issues a warning if the optimized regularization parameter λ exceeds the empirical threshold of $\lambda > 0.1$, which we have observed in signal-free cases. We also recommend users examine the denoised features: TV denoising will not reveal signals that were previously unobservable; rather, it improves interpretability by suppressing background noise, allowing real but weak features – i.e. previously visible only at low contour levels – to emerge more clearly.

3. Following the argument on lines 115-117 and line 125-126, to what extent does the method, conversely, “simply” act like a low-pass filter, suppressing high-frequency features in the difference map that are less likely to be true signal? This question has a few parts: (a) would a conventional low-pass filter (e.g., Gaussian) be able to achieve the same effect? (b) is there a case to be made that some type of low-pass filtering is indeed desirable and that the TV approach is better than others? I do not know the relevant literature in image analysis well, and neither will most readers. It would be helpful to bring us some insight; (c) to what extent are true sharp spatial features likely to be lost?

Below, we respond to each sub-question.

(a+b) First, let us start with the theoretical case for TV denoising over common alternatives. Linear filters, such as Gaussian smoothing, reduce high-frequency content uniformly across the image. While

these filters are computationally efficient and easy to implement, they blur both signal and noise. This approach assumes that features that vary rapidly across space – such as sharp edges or small peaks (i.e., high-frequency/resolution components) – are likely to be noise, not signal. However, DED maps may contain sparse, spatially localized features across a broad frequency range. These features are often critical for resolving subtle but meaningful structural changes and should be preserved if at all possible.

By contrast, TV denoising assumes that the underlying signal is predominantly piecewise smooth but allows for sharp transitions. This made TV denoising our candidate of choice for DED map enhancement, particularly when the map consists of broad flat regions interrupted by sparse, high-gradient features embedded in noise.

The reviewer is correct, however, that theory is just theory, and we should show benchmarks of TV denoising against simple, common denoising filters. In our revised manuscript, we report the performance of TV denoising to Gaussian and median filtering on our benchmark CI-rsEGFP2 DED map. The TV-denoised map appeared with visually stronger and more interpretable chromophore signals:

We have added this to the Supplement as Figure S14.

This qualitative assessment based on map visualization was supported by higher negentropy values for the TV-denoised maps as compared to Gaussian or median filtering.

Further, in our introduction on page 7, we have added a reference to the review by Fan et al. (2019) [new reference 31], which offers a clear classification and discussion of the strengths and limitations of various methods across image processing contexts.

(c) As discussed in (a + b), TV denoising is designed to suppress noise while preserving sharp, localized features – a behavior that makes it particularly well suited to DED maps. Like any denoising method, it may still suppress weak but genuine signal if the regularization is too strong. To reduce this risk, we select the parameter automatically by maximizing negentropy, which favors retention of structured, non-Gaussian features.

In our experience, this strategy consistently preserves relevant signal, even in low signal-to-noise regions. Some loss of very weak features remains theoretically possible, but in practice, our tests showed better signal retention compared to Gaussian or median filtering (new Figure S14).

We also distinguish this from the separate concern of producing spurious features in signal-free maps, which we address in our response to the reviewer’s comment on negative controls (and in the manuscript, p. 19–20). There, we describe the “staircasing” artifact associated with TV denoising in flat regions and explain how METEOR flags such cases based on unusually high λ values.

4. Although the software is, laudably, open-source, the authors do not provide access to their maps. Could they do so, e.g., through Zenodo or another repository?

We agreed that this was a good suggestion. We have generated a Zenodo repository where we uploaded the different denoised maps shown in the paper.

Page 26 addition: *The code is available at: <https://github.com/rs-station/meteor>. Map files from TV denoising examples shown are available at: <https://zenodo.org/records/15800905>.*

Technical points:

1. Negentropy: it is not uncommon for structure factor amplitudes for different conditions to vary in average size after initial data processing. This has been addressed by various local scaling approaches implemented in, e.g., SOLVE, SCALEIT, and in estimation of the local Σ in implementations of the French-Wilson algorithm (e.g. in PHENIX). Without such local scaling, features of the density may contaminate the difference density. I could imagine such artifacts increasing negentropy but for the wrong reasons. It would be helpful for the authors to introduce a note of caution about the use of negentropy as a measure of difference map quality if they agree this concern is justified (referred to as such in lines 99, 106).

2. Outliers: Line 155-156: similar to comments above about imbalance between datasets, the invocation of the CLT suggests that meteor may not be robust to outlier Delta F’s and that k weighting, which suppresses outliers, may improve the reliability of meteor. Interestingly (lines 172-177), the authors use negentropy to tune k at least in one case. That seems risky if true outliers are present as I could imagine them increasing negentropy if not suppressed.

We agree completely – there are important potential pitfalls when using negentropy as a map quality metric. In principle, both uncorrected amplitude scaling and outliers in the data could lead to misleadingly high negentropy values due to spurious, non-biologically relevant signals. Any heuristic metric, including negentropy, is susceptible to pathologies.

That said, as used by us across a broad range of test cases (including new cases by external users since our initial submission) we have not yet encountered a case where negentropy optimization led to qualitatively incorrect conclusions. In fact, as we show in Figures 2 and S2, negentropy tracks well with human intuition and successfully guides the selection of both k-weighting and TV regularization parameters across reasonable ranges.

It's important to note, however, that unconstrained optimization of negentropy would be very unwise. For instance, a map with all density concentrated in a single voxel would produce a high negentropy but minimal interpretability or biological value. This highlights that negentropy should only be used to discriminate between physically plausible possibilities and cannot be used in an unconstrained manner.

To address this concern explicitly, as the reviewer suggested, we have a cautionary paragraph at the beginning of the discussion:

Page 18: To this end, we find negentropy to be an effective reporter of difference map quality when seeking optimal amplitude weighting and denoising parameters. Negentropy maximization alone, however, cannot be used for map denoising without other restrictions to the procedure. Consider a putative DED map with half of the voxels randomly set to +x and half to -x. Such a map will have a high negentropy, but no information content. Instead, negentropy, which reports how noise-like a signal is, should be thought of as a measure of the signal-to-noise ratio in a map. It can tell if the map contains signal, but not if that signal is faithful to any crystallographic or biological reality. Therefore, negentropy should only be used to evaluate competing DED maps generated by methods that can maintain fidelity to the original crystallographic data.

We have also added a more advanced discussion of systematic errors and references to relevant methods:

Page 19: It is important to note that METEOR, as described throughout this work, is designed to address Gaussian noise in DED maps and does not inherently correct for systematic errors common in crystallographic datasets, such as scaling artifacts or anisotropy. To improve the reliability of denoised maps, we recommend applying tools specifically developed to correct for such systematic effects as part of standard preprocessing prior to TV denoising. For example, AIMLESS adjusts the measured reflection intensities to correct for systematic errors and variations – such as changes in beam intensity, or detector sensitivity – so that all measurements are placed on a common, consistent scale for accurate merging and STARANISO applies direction-dependent scaling corrections and resolution

cutoffs to address anisotropic diffraction; SCALEIT offers local scaling procedures that mitigate systematic variation in structure factor amplitudes. These approaches are well-established in difference map workflows and are complementary to the denoising strategies we present here.

Confusingly, line 429 also describes a k-weighting step during the iterative TV version of the algorithm without providing further detail. How? Is it important?

We should clarify: k-weighting is not part of the iterative denoising and phase estimation loop itself. Rather, it is applied in a single post-processing step after the final iteration, prior to the final denoising and output map generation. We agree that the original phrasing is confusing and have now revised the Methods section to be more precise:

After the final iteration, by default meteor applies k-weighting and a final round of TV denoising as a post-processing step to produce the output map.

Minor points

The authors may want to refer to METEOR in their title or abstract. This would not be the first crystallographic software for which the corresponding paper would be hard to find.

We thought about this suggestion and decided we would prefer not to include an unexplained acronym in the title. However, the reviewer's point was good in terms of ensuring the method is more easily discoverable: we have now added a mention of the METEOR package in the abstract. Upon reflection, we have also revised the title to more deeply reflect the core methods of the paper – now reading: *“Denoising and Iterative Phase Recovery Reveal Hidden States in Protein Crystals.”*

Notably, the authors refer to <30% occupancy as low (line 75) and then apply their method in Figure 4 and lines 252-256 to ligands of higher occupancy. It would be more interesting to see the effect on difference maps of lower-occupancy ligands.

It is good that the reviewer highlighted this inconsistency. Upon reflection, we agree that our original statement (line 75) referring to occupancies below 30% as a benchmark was arbitrary and did not effectively communicate the challenges we sought to address.

Our goal in selecting examples was not strictly to demonstrate performance as a function of occupancy, especially with a strict cutoff, but rather to highlight cases where interpreting the difference density maps with existing methods is especially difficult. Often such cases occur when a minor state is present with a relatively small occupancy, but occupancy is only one factor that can reduce signal-to-noise: experimental noise, structural heterogeneity, lattice disorder, and ligand flexibility will all play a role.

The time-resolved CI-rsEGFP2 dataset and the ligand-binding cases in Figure 4 present maps that are challenging to interpret due to weaker and noisy signals. We think these are good examples to show how METEOR meaningfully improves interpretability compared to conventional methods. We have revised the original statement highlighted by the reviewer to more accurately reflect this nuance – the challenge lies in the detectability of signal in difference maps, where partial occupancy and noise are important contributing factors.

Combined with measurement noise, this partial occupancy means that dataset-dependent changes can be small, making DED map features difficult to interpret.

Line 88: “this manual intervention requires extensive trial-and-error ...” – to my limited knowledge, such optimization is not commonplace, but I could be wrong.

In our own experience as users in the field, and especially when helping others work through challenging cases, we have often seen that groups spend a lot of time tweaking these parameters in the hope of observing signals they anticipate. This “optimization” is not always documented but it happens and can, naturally, affect conclusions.

There are several instances of residual text editing (lines 140, 172, 221, 319, ...)

Thank you very much for pointing these out. We’ve made sure they are all removed in the new version of the manuscript.

Line 142: “are compatible and stackable with existing methods” – is there any potential for complications the reader should worry about? Are there any particularly interesting cases of stacking where meteor may complement another approach well?

We do not anticipate any complications when combining METEOR with existing methods, as our approach modifies only the map (or map-derived structure factors) without altering the underlying data or models. Following the second point of this comment, we have revised the text to specify that METEOR is particularly well suited for integration with tools designed for occupancy estimation (which METEOR currently does not try to address in a novel way). We have found METEOR to be especially helpful in cases where signal is near the noise floor and more traditional difference maps are difficult to interpret.

Page 8: Importantly, our observations on using map negentropy and TV denoising to improve difference density signals are compatible and stackable with existing methods. In particular, METEOR can complement existing occupancy estimation and event detection tools – such as PanDDA or Xtrapol8 – by improving map interpretability prior to modeling. We expect this stacking to be especially useful in low-occupancy cases where difference map signal is weak.

Figure S1: The description of n as the “noise fraction” is highly confusing. Reading the supplement, it becomes clear that n is the fraction of pixels in the true difference density map with negligible difference density, rather than something like $1 - n$ (excited state fraction). Going by the examples in Fig 1a-c, I would doubt whether any scenario with $n < 0.99$ or so is ever relevant in practice! The example in Figure S1 feels rather contrived. Perhaps the authors could take the histograms in Figure 1a-c, all of which have a dominant central mode in their noise-free DED histogram, and add synthetic noise as in Fig S1. I would be surprised if, in such an exercise, the kurtosis would exhibit non-monotonic dependence on n (or more importantly, σ_n !).

We found this to be a very good suggestion and agreed that it would be a more intuitive way of testing the behavior of different metrics. We have therefore replaced the old Figure S1 with a new version where synthetic noise is added to the examples in Figure 1a-c. We have also updated the introduction of this figure in the main text and legend:

Page 9: Skewness, kurtosis, and negentropy are well-known measures for non-Gaussianity. In addition to negentropy, described above, skewness measures asymmetry in a distribution, while kurtosis describes the "tailedness" or sharpness of its peak. To investigate the suitability of these statistics as indicators of difference map quality, we extend the examples from Figure 1(a-c) by adding Gaussian noise to the calculated real-space signal (see Figure S1). We introduce increasing levels of noise and find that negentropy decreases monotonically with the addition of noise and is the most robust in its behavior when compared to skewness and kurtosis (Figure S1). On the basis of this test, we proceed with the proposal that negentropy could be a useful metric to evaluate the signal-to-noise ratio in difference density maps.

Line 211: “additive noise” --- added in real space or reciprocal space?

As a simple test case, we add noise in real space to the calculated difference map. We have now specified this in the text. Note that since the Fourier transform is linear, every case of additive noise in real space can be achieved by adding noise in reciprocal space and *vice versa*, though one has to be sure to get the scale (units) correct. We also tested adding noise to the structure factor differences in reciprocal space and found that METEOR was equally successful in denoising.

Figure 2b, power spectrum: it would be nice to have a log-spaced x-axis like in 2a as most of the interesting information, at high resolution, is essentially uninterpretable in the current display. It is also hard to tell apart the colors in these power spectra.

In response to this comment and a note by Reviewer 1, we’ve made the labeling and coloring of the power spectra in Fig 2 clearer. For Figure 2b, we have now added a secondary axis for resolution, so it is easier to relate resolution to d^{-2} at a glance. We hope the text we added to page 11, which now

introduces and describes the concept of the power spectrum, and these additional edits to the figure legends aid the reader.

Figure 3: What are the units of phase change? Confusingly, the caption refers to 3c showing the mean (absolute?) change in phase, while the main text (line 237) speaks of cumulative phase change—which, I would assume, goes up rather than down. At the same time, if this is the phase change per iteration, the cumulative change would seem very large if the units are radians rather than degrees! Also welcome would be a measure of deviation from an Fc–Fc map (real-space correlation or, e.g., $\langle \cos(\Delta\phi) \rangle$).

The y-axis in Figure 3c plots the **average absolute phase difference in degrees**, not a cumulative quantity. We have now corrected the y-axis label and the corresponding sentence in the main text (line 237), which was a leftover from an earlier draft version of the plot. Thank you for catching this!

Regarding the suggestion to include a measure of deviation from an Fc–Fc map, we agree this is very useful and have included it in the Supplementary Information (Figure S3) for a synthetic dataset where the ground truth is known. However, for the experimental case in Figure 3, we cannot show $\langle \cos(\Delta\phi) \rangle$ or real-space correlation to Fc–Fc maps, since the true phases are unknown.

Line 364: “... potentially doubling ...” – I can see why the authors might say this but it seems like an exaggeration at present.

This is a good observation. Upon re-reading, we agree that the phrasing “potentially doubling” may overstate the possible results. We have revised the sentence to clarify that, in theory, recovering the perturbed-state phases could unlock up to a twofold improvement in signal-to-noise, and that our method aims to approximate this ideal in practice:

Page 18: *We hypothesized that by extending single-pass total variation denoising (single-TV) into an iterative map improvement algorithm (it-TV), it would be possible – at least in theory – to recover phase information and recover a fraction of the signal-to-noise loss relative to traditional $F_{obs} - F_{obs}$ maps that rely on the native-phase approximation.*

Line 370-371: is this a relevant concern? If so, when?

This is a fair question in response to our statement in the text. While local minima are a known theoretical limitation of iterative projection algorithms like Gerchberg–Saxton, in our experience they have not posed a practical problem, likely because as compared to other applications of Gerchberg–Saxton (e.g. *de novo* phasing in CDI) we start with a very good phase estimate from the native model. The theoretical concern is that, in cases with particularly high noise or extremely weak signal, the iterative procedure may converge prematurely to an incorrect minimum. We have modified the original text to further clarify this point:

Page 18: *Our current it-TV implementation, based on the Gerchberg-Saxton iterative projection algorithm, achieves this improvement effectively; while in principle such algorithms can become trapped in local minima, we have not observed clear failure cases in our test data so far. Nonetheless, this could remain a consideration for more challenging datasets and motivates future work on more robust or regularized update schemes, such as Feinup’s hybrid-input output or its descendants, to enhance it-TV robustness against noise-induced errors.*

Line 376-377: what is the statement based on that Fo-Fo maps “contain weaker signals” than “Fo-Fc” maps (are modeling errors signals?)?

This was poorly phrased, thank you for catching it. With “contain weaker signals” we simply meant that our Fo-Fo example is the lowest occupancy of the ones shown. We have improved the phrasing in the relevant section on page 19.

The largest gains from our TV denoising tests occur for the Cl-rsEGFP2 $F'_{obs}-F_{obs}$ map, which is expected to benefit the most from the replacement of the single-phase approximation in it-TV and also represents the lowest occupancy example. We therefore anticipate that our denoising methods will be the most helpful in pushing the boundary for interpreting datasets from low-occupancy time-resolved studies or ligands that are weakly bound.

Lines 419-429: It would be helpful to guide the reader more explicitly through Figure S6. This is a key figure but hard to digest

We re-worded the figure legend for Figure S6 and added new details in the section on page 23 to be more clear and helpful:

Native (F_{obs}) and derivative (F'_{obs}) state amplitudes are provided as input, together with a reference model. A difference map is computed using the phases calculated from the reference model and the difference structure factor amplitudes ($F'_{obs} - F_{obs}$). This starting difference map is denoised using Chambolle’s total variation algorithm with a λ value optimized via negentropy. The resulting denoised map is inverse Fourier-transformed to obtain a set of complex structure factor differences, ΔF^{TV} . We then project these onto the constraint circle defined by the derivative-state amplitudes ($|F'_{obs}|$) to estimate new phases ϕ^{proj} (Figure S6). This projection aims to provide improved phase estimates for the derivative dataset. Using these refined amplitudes and projected phases, we compute a new difference map, apply TV denoising again, and repeat this procedure until convergence – defined as when the mean absolute phase change between iterations drops below 10^{-3} degrees. By default, METEOR then applies k -weighting and a final round of TV denoising as post-processing.

Line 480: pandas, not panda?

Thank you for spotting this – corrected.

Lines 629-633: “from a standard normal distribution” – of the same variance?

In the normal probability plots we show, the theoretical quantiles are drawn from a standard normal distribution with mean 0 and unit variance (as is standard practice). The reviewer is correct in pointing out that this was not clear and should be specified: we have revised the sentence to make this explicit.

Page 34: (...) the observed data (sample quantiles) are ordered and plotted against the expected values of the ordered statistics for a sample from a standard normal distribution (mean 0, variance 1) of the same size as the data (theoretical quantiles) (...)

Figure S8b: the legend seems to cover the interesting part of the graph.

Thank you for pointing this out, we moved the legend away from the peak.

REVIEWER 3

The manuscript "Denoising Reveals Low-Occupancy Populations in Protein Crystals" by Professor van Thor and colleagues [Reference COMMSBIO-24-8699-T], presents an innovative framework that integrates negentropy and total variation (TV) denoising to enhance difference electron density (DED) maps, addressing key challenges in low-occupancy signal detection for structural biology. This methodology represents a significant advancement in the field by introducing negentropy as a robust metric to optimize difference map parameters, offering an automated and reproducible alternative to traditional trial-and-error approaches. Furthermore, the iterative-TV (it-TV) approach is a notable development, allowing for iterative phase refinement to improve the interpretability of weak signals. These innovations are complemented by validation using both synthetic and experimental datasets, demonstrating the broad applicability of the method in diverse contexts such as time-resolved crystallography and fragment screening.

The authors skillfully justify their choice of negentropy as a measure of map quality, showing that it effectively quantifies non-Gaussianity in voxel value distributions and is superior to other metrics like skewness and kurtosis. The integration of TV denoising significantly improves the signal-to-noise ratio in DED maps, as demonstrated in both single-pass and iterative applications. Importantly, the manuscript highlights the practical advantages of their approach, including its accessibility as an open-source package (METEOR) and its computational efficiency, making it suitable for real-time experimental workflows and high-throughput campaigns.

The authors' innovative use of negentropy and TV denoising addresses longstanding challenges in the analysis of low-occupancy states, offering a broadly applicable solution that is particularly relevant for time-resolved and fragment-screening crystallography. The figures effectively illustrate the method's capabilities, with clear improvements in signal clarity and interpretability across all test cases. For example, the application of it-TV to datasets involving Cl-rsEGFP2 photoisomerization and fragment-bound SARS-CoV-2 main protease provides striking evidence of the method's ability to reveal chemically meaningful features that were previously obscured by noise.

However, there are some areas that would benefit from further discussion.

1) While the authors acknowledge the limitations of the it-TV approach in high-noise conditions due to potential susceptibility to local minima, they could provide a more detailed discussion of how these challenges might be mitigated in future developments. Additionally, the manuscript primarily addresses Gaussian noise and does not explicitly consider the effects of systematic errors common in crystallographic datasets, such as scaling artifacts or anisotropy. While these limitations do not detract from the immediate utility of the method, acknowledging them more explicitly and suggesting potential solutions or complementary approaches would strengthen the manuscript.

We thank the reviewer for this constructive comment and agree with the proposed direction. We have expanded the discussion to more explicitly acknowledge METEOR's sensitivity to systematic errors such as scaling artifacts and anisotropy. We have also added specific suggestions for complementary tools and preprocessing steps – such as AIMLESS, STARANISO, and SCALEIT – that can be used alongside TV denoising to address these issues in practice:

Page 19: It is important to note that METEOR, as described throughout this work, is designed to address Gaussian noise in DED maps and does not inherently correct for systematic errors common in crystallographic datasets, such as scaling artifacts or anisotropy. To improve the reliability of denoised maps, we recommend applying tools specifically developed to correct for such systematic effects as part of standard preprocessing prior to TV denoising. For example, AIMLESS adjusts the measured reflection intensities to correct for systematic errors and variations – such as changes in beam intensity, or detector sensitivity – so that all measurements are placed on a common, consistent scale for accurate merging and STARANISO applies direction-dependent scaling corrections and resolution cutoffs to address anisotropic diffraction; SCALEIT offers local scaling procedures that mitigate systematic variation in structure factor amplitudes. These approaches are well-established in difference map workflows and are complementary to the denoising strategies we present here.

2) Another point of consideration is computational scalability. While the authors describe their approach as computationally efficient, more specific information about runtime performance across datasets of varying sizes would provide valuable context, particularly for users planning to apply this method in high-throughput scenarios.

TV denoising is computationally inexpensive: on a single core, a typical TV run will complete in around 1 minute, while it-TV will complete in around 3 minutes.

We have included the plot below in the SI and referenced it in our Discussion. This shows run times for the full single-pass TV denoising protocol of the P212121 Cl-rsEGFP2 dataset (unit cell dimensions: 51.99 62.91 72.03 90.00 90.00 90.00, resolution: 1.6 Å). A map sampling rate of 4, as defined by GEMMI/CCP4 is standard.

3) Furthermore, while the manuscript effectively demonstrates the method's strengths relative to existing tools like PanDDA and Xtrapol8, a more in-depth qualitative comparison with emerging deep learning-based denoising techniques would help contextualize the advantages of TV denoising.

We have expanded the section to include a more detailed comparison between our TV-based denoising approach and emerging deep learning methods. While deep learning could be a promising future direction for DED map analysis – and one that could ultimately complement or enhance methods like METEOR – we believe our current approach provides a practical, and easily implemented solution for present-day use, particularly in cases where generalizable training data are limited or unavailable (as is the case for time resolved datasets, for example). We now reference recent deep learning efforts and clarify the relative advantages and trade-offs of both approaches in the main text:

Page 20-21: (...) *Future developments in DED map denoising could focus on a more holistic approach. For example, recent advances in deep learning, particularly convolutional neural networks (CNNs) and diffusion-based architectures, have shown strong performance in image restoration tasks and are beginning to find applications in structural biology, including electron density map enhancement [new references added, 50-52]. These models can learn complex, data-driven priors and may be capable of distinguishing subtle signal from structured noise or artifacts, even in low signal-to-noise regimes. However, current deep learning approaches often require large, well-annotated training datasets, which, for example, are not readily available when dealing with DED maps. By contrast, our TV-based denoising framework is data-agnostic and easily deployable, requiring no training phase. This makes it equally suited to both fragment screening datasets and time-resolved experiments, including those that capture rare or previously unobserved conformational changes. TV denoising is fast, computationally inexpensive, and can be run from most workstations without the need to train an additional set of parameters.*

Despite these considerations, the manuscript is a strong candidate for publication in Communications Biology. Its methodological innovations are timely and impactful, addressing critical needs in the field. While additional benchmarking or testing under non-Gaussian noise conditions could further enhance the work, these aspects can be reasonably addressed through expanded discussion rather than new experiments. Overall, this study advances the state of the art in DED map analysis and is likely to have a lasting impact on both methodological development and practical applications in structural biology.

Thank you for the encouraging assessment of our work. We have worked to expand the Discussion to address the points raised. We also hope that the additional data and explanations provided in response to Reviewer 2 and to benchmark computation costs/time are helpful in advancing the scope of the manuscript.